# Spatiotemporal dynamics of SETD5-containing NCoR–HDAC3 complex determines enhancer activation for adipogenesis

Yoshihiro Matsumura [1,12✉], Ryo Ito[2,12], Ayumu Yajima[1,3,12], Rei Yamaguchi[2], Toshiya Tanaka [4], Takeshi Kawamura[5], Kenta Magoori [1], Yohei Abe [1], Aoi Uchida[1], Takeshi Yoneshiro [1], Hiroyuki Hirakawa[1,6], Ji Zhang[1,2], Makoto Arai[1,2], Chaoran Yang [2], Ge Yang[2], Hiroki Takahashi[2], Hitomi Fujihashi[1], Ryo Nakaki[7,8], Shogo Yamamoto [7], Satoshi Ota[7], Shuichi Tsutsumi[7], Shin-ichi Inoue [2], Hiroshi Kimura [9], Youichiro Wada[5], Tatsuhiko Kodama[4], Takeshi Inagaki [1,10], Timothy F. Osborne[11], Hiroyuki Aburatani [7], Koichi Node [3] & Juro Sakai [1,2✉]

Enhancer activation is essential for cell-type specific gene expression during cellular differentiation, however, how enhancers transition from a hypoacetylated "primed" state to a hyperacetylated-active state is incompletely understood. Here, we show SET domain-containing 5 (SETD5) forms a complex with NCoR-HDAC3 co-repressor that prevents histone acetylation of enhancers for two master adipogenic regulatory genes *Cebpa* and *Pparg* early during adipogenesis. The loss of SETD5 from the complex is followed by enhancer hyperacetylation. SETD5 protein levels were transiently increased and rapidly degraded prior to enhancer activation providing a mechanism for the loss of SETD5 during the transition. We show that induction of the CDC20 co-activator of the ubiquitin ligase leads to APC/C mediated degradation of SETD5 during the transition and this operates as a molecular switch that facilitates adipogenesis.

[1] Division of Metabolic Medicine, Research Center for Advanced Science and Technology, The University of Tokyo, Tokyo, Japan. [2] Division of Molecular Physiology and Metabolism, Tohoku University Graduate School of Medicine, Sendai, Japan. [3] Department of Cardiovascular Medicine, Saga University, Saga, Japan. [4] Department of Nuclear Receptor Medicine, Laboratories for Systems Biology and Medicine, Research Center for Advanced Science and Technology, The University of Tokyo, Tokyo, Japan. [5] Isotope Science Center, The University of Tokyo, Tokyo, Japan. [6] Department of Physiology and Cell Biology, Tokyo Medical and Dental University (TMDU), Graduate School, Tokyo, Japan. [7] Genome Science Division, Research Center for Advanced Science and Technology, The University of Tokyo, Tokyo, Japan. [8] Rhelixa Inc, Tokyo, Japan. [9] Cell Biology Center, Institute of Innovative Research, Tokyo Institute of Technology, Yokohama, Japan. [10] Laboratory of Epigenetics and Metabolism, Institute for Molecular and Cellular Regulation, Gunma University, Gunma, Japan. [11] Institute for Fundamental Biomedical Research, Johns Hopkins All Children's Hospital, and Medicine in the Division of Endocrinology, Diabetes and Metabolism of the Johns Hopkins University School of Medicine, Petersburg, FL, USA. [12] These authors contributed equally: Yoshihiro Matsumura, Ryo Ito, Ayumu Yajima. ✉email: matsumura-y@lsbm.org; jmsakai@med.tohoku.ac.jp

Cell-type specific gene expression is spatially and temporally regulated by DNA cis-regulatory elements called enhancers that can be located at varying locations relative to their target genes ranging from a few hundred to mega-base distances. Enhancers interact with proximal promoter elements via long-range chromatin looping events mediated by transcription factors that are recruited to specific binding sites located within both enhancers and proximal promoters through systematic interactions with non-DNA binding transcriptional coregulatory and chromatin modifying complexes and the combined action results in regulated augmentation of basal transcription levels[1]. Enhancers are selected and activated in a cell-type specific manner during development and in response to specific stimuli. Enhancer selection and activation involve binding of lineage-determining transcription factors to their recognition motif combined with local nucleosome remodeling[2–4].

The transcription factor complexes that bind to enhancer and promoter motifs contain histone modification enzymes that mark enhancer and proximal promoter regions with specific histone modification signatures. For example, enhancer chromatin displays an enrichment for histone H3 lysine 4 mono- methylation (H3K4me1) and low levels of H3K4 tri-methylation (H3K4me3) compared with promoters which show enrichment for H3K4me3[5]. Enhancers can be classified as poised, primed, or active, defined by a combination of unique histone modifications which can be interconverted[6] (also reviewed in[4]). Poised enhancers are marked with H3 lysine 27 tri-methylation (H3K27me3), a repressive epigenetic mark (see also Fig. 1a). Poised enhancers are inefficient at driving gene expression, however, they are converted to active enhancers during differentiation through the loss of repressive H3K27me3 and a gain of the activating H3 lysine 27 acetylation (H3K27ac) mark. Prior to activation, enhancers can also exist in a primed state. Primed enhancers are characterized by the presence of H3K4me1 but they lack both repressive H3K27me3 and active H3K27ac. Additional cues such as signal dependent stimuli result in recruitment of additional transcription factors, and the eventual recruitment of co-activators leading to enhancer activation. Active enhancers simultaneously contain both H3K4me1 and H3K27ac.

The acetylation state of a given chromatin locus is controlled by two classes of antagonizing histone modifying enzymes, histone acetyltransferases (HATs; e.g., CBP and p300)[5,7] and deacetylases (HDACs), which are components of co-repressors such as nuclear receptor co-repressors (NCoRs) and SMRT[8–10]. HATs facilitate H3K27 acetylation[5,7], whereas HDAC-containing complexes remove H3K27 acetylation[11,12]. Because H3K27ac determines whether an enhancer is primed or active[6], the relative recruitment ratio of HATs "writers" and HDACs "erasers" to enhancer chromatin is likely a key determinant for acetylation status. However, molecular mechanisms that keep enhancers in a primed state are not fully understood.

The differentiation program converting preadipocytes into adipocytes has been well studied, especially in cultured mouse 3T3-L1 cell lines[13,14]. Growth-arrested 3T3-L1 preadipocytes synchronously reenter the cell cycle and undergo two rounds of mitotic clonal expansion followed by acquisition of the mature adipocyte phenotype[15]. This process, called adipogenesis, is orchestrated by a cascade of sequentially acting cell cycle proteins, transcription factors, and chromatin modifiers that shape the differentiation through the symphonic interplay of hormones and other signaling pathways (reviewed in[13,16–18]). During this period, there is a rapid and transient induction of transcription factors CCAAT/enhancer-binding protein β (C/EBPβ) and C/EBPδ (2–4 h), that are recruited to binding sites of C/EBPα and PPARγ gene enhancers (4–24 h), yet they remain transcriptionally silent. After 36 h, when the first round of mitosis is completed during clonal expansion, enhancers are transformed to the active state which is followed by gene activation (36 h and later)[19,15] (also reviewed in[18]). These two master regulators, C/EBPα and PPARγ, together orchestrate the global changes in gene expression that cause the acquisition of the mature fat-laden adipocyte phenotype as the second round of mitosis is completed[15,20,21].

The anaphase-promoting complex/cyclosome (APC/C) is an evolutionarily conserved multi-subunit ubiquitin ligase that controls cell cycle progression through proteasomal degradation of key cell cycle regulatory targets[22]. Recent studies have revealed it also plays a role in postmitotic cellular responses to developmental signals through transcriptional regulation of genes by directly targeting various cell type-specific transcription factors and their regulators for degradation. The catalytic core of APC/C is comprised of the cullin subunit ANAPC2 and RING domain subunit ANAPC11. CDC20 and CDH1, two structurally homologous accessory subunits, are referred to as "APC/C coactivators" and they mediate substrate recognition and stimulation of the ubiquitin ligase activity of APC/C. CDC20 binds and activates APC/C in M phase to trigger chromatid separation and mitotic exit (reviewed in[23], see also the schematic illustration in Supplementary Fig. 3c).

Our previous genome-wide search for epigenetic regulators involved in adipogenesis revealed several genes encoding SET domain-containing proteins[21]. Among them, SETD5 is a potential histone methyltransferase whose gene expression declines during the first 48 h of adipogenesis[21]. Recent genetic studies reported loss-of function mutations in SETD5 in patients with intellectual disability[24–26] and studies in Setd5 mutant mice suggested roles of SETD5 in neuronal development, however, whether it actually contains histone methyltransferase activity has been controversial and could be context dependent[27–30].

In this study, we show that SETD5 performs a structural role in formation of a complex with NCoR-HDAC3 repressor complex to keep enhancers hypoacetylated and in a "primed" state by restricting recruitment of HATs to the enhancers of Cebpa and Pparg genes. Degradation of SETD5, which is mediated by APC/C E3 ubiquitin ligase, facilitates HAT recruitment to primed enhancers for hyperacetylation. Thus, the spatiotemporal dynamics for SETD5 residence in NCoR-HDAC3 co-repressor complex modulates enhancer transition from the primed to active state that is required for terminal differentiation of adipocytes.

## Results

**Primed enhancers of adipogenic master genes become active during the early phase of adipogenesis.** To understand the regulation of chromatin modifications associated with enhancer activity during adipogenesis, we analyzed repressive H3K27me3, permissive H3K4me1, and active H3K27ac by chromatin immunoprecipitation sequencing (ChIP-seq) during differentiation of cultured 3T3-L1 preadipocytes. Consistent with prior studies, C/EBPβ and C/EBPδ which are early pro-adipogenic transcription factors[16,31], were expressed in preadipocytes but their transcript levels were further induced during the first 48 h of differentiation (Supplementary Fig. 1a). Mandrup and her colleagues, using the Hi-C technique, previously identified looping between Cebpb promoter and regions located at +77 kb and +88 kb[32]. As shown in Supplementary Fig. 1b, we also observed that these Cebpb distal regions (e.g. +77 kb and +88 kb) and also Cebpd distal regions (e.g. +62 kb and +98 kb) displayed active enhancer signatures marked by the presence of both H3K4me1 (track 3) and H3K27ac (track 4) which was coupled with a paucity of H3K27me3 (track 2) in preadipocytes before

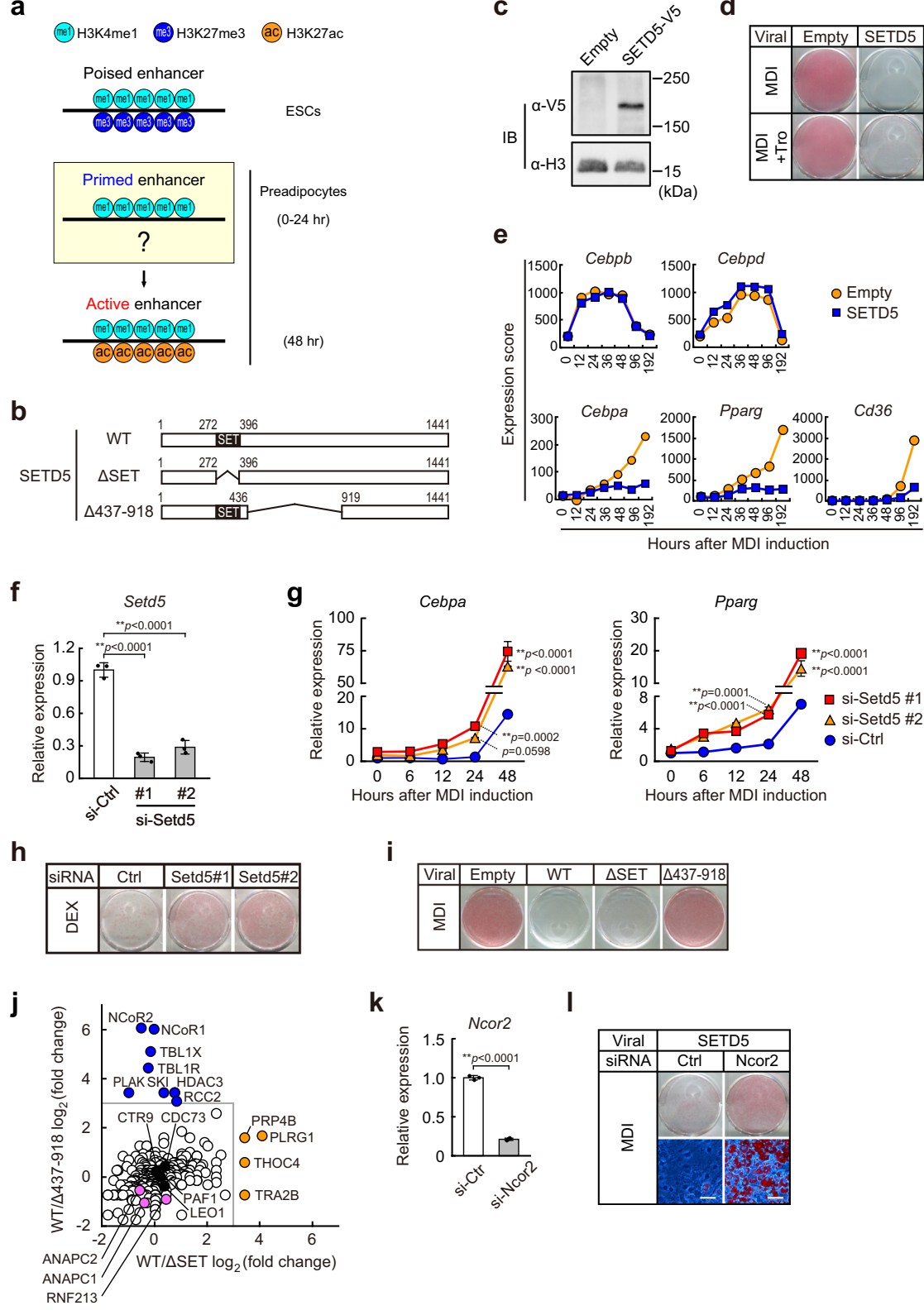

differentiation (i.e. 0 h) indicating that these enhancers are already active before differentiation. Information in the recent data base Enhancer Atlas 2.0 showed that both *Cebpb* and *Cebpd* distal regions are annotated as enhancers in 3T3-L1 cells (Supplementary Fig. 1c)[33]. By contrast, C/EBPα and PPARγ, both of which are late-adipogenic transcription factors[16], were expressed at very low levels early but they were robustly induced between 24

and 48 h of differentiation (Supplementary Fig. 1a). Supplementary Fig. 1d shows that before differentiation (0 h), genomic regions distal to *Cebpa* (e.g. +8 kb, +19 kb, and +53 kb) and *Pparg* (e.g. +60 kb and +158 kb) had H3K4me1 (track 3) but lacked H3K27me3 (track 2) and H3K27ac (track 5), indicating that these enhancers were in a "primed" state[34]. These *Cebpa* and *Pparg* distal regions are annotated as enhancers in 3T3-L1 cells in

**Fig. 1 SETD5 associates with NCoR and HDAC to inhibit *Cebpa* and *Pparg* induction during adipogenesis. a** Schematic illustration of enhancer states. Poised enhancer in embryonic stem cells (ESCs) is characterized by the presence of H3K4me1 and H3K27me3. Primed enhancer is characterized by the presence of the only H3K4me1, while active enhancer is characterized by the presence of both H3K4me1 and H3K27ac[6]. **b** Schematic representation of mouse SETD5 protein and its mutants. **c** Retroviral expression of SETD5 in 3T3-L1 preadipocytes. Nuclear proteins from indicated 3T3-L1 preadipocytes were subjected to immunoblot (IB) analysis. Equal loading of the proteins was confirmed by blotting with anti-histone H3 (H3) antibody. **d** ORO staining of 3T3-L1 preadipocytes transduced with empty virus or SETD5. Preadipocytes were induced with MDI mixture or MDI mixture plus troglitazone (Tro). **e** Transcriptional changes of adipogenic genes in SETD5-transduced 3T3-L1 preadipocytes during adipogenesis by using a microarray. **f, g** siRNA-mediated knockdown of SETD5 in 3T3-L1 preadipocytes. *Setd5* (**f**), *Cebpa*, and *Pparg* (**g**) mRNA expression was quantified by qPCR. **h, i** ORO staining of SETD5 knocked-down 3T3-L1 preadipocytes (**h**) or 3T3-L1 preadipocytes transduced with empty virus, SETD5, or mutants (**i**). Preadipocytes were induced with Dex (**h**), or MDI mixture (**i**). **j** Proteomics analysis of SETD5 interacting proteins. Nuclear proteins from WT-, ΔSET-, and Δ437-918-SETD5-transduced preadipocytes were subjected to immunoprecipitation with anti-V5 antibody followed by trypsin digestion and mass spectrometry. Color code: proteins lost (<2$^{-3}$-fold) by deletion of a.a. 437–918 (blue); proteins lost by deletion of SET domain (orange); proteins not affected by deletion of a.a. 437–918 or SET domain (white). Proteins of PAF1 complex are highlighted in black. Proteins of APC/C complexes and RNF213 are highlighted in pink. **k** siRNA-mediated knockdown of NCoR2 in SETD5-transduced 3T3-L1 preadipocytes. *Ncor2* mRNA expression was quantified by qPCR. **l** ORO staining of NCoR2 knocked-down, SETD5-transduced 3T3-L1 preadipocytes. Scale bar = 50 μm. **c, f, g, k, l** Representative of three independent experiments. **f, g, k** Data are mean ± SD of three technical replicates. **f, g** One-way ANOVA with Tukey's multiple comparisons test. **k** Unpaired two-tailed Student's *t*-test. **p < 0.01. Source data are provided as a Source data file.

Enhancer Atlas 2.0 (Supplementary Fig. 1e)[33]. *Pparg* gene uses an alternative transcription start site (+60 kb) to produce *Pparg2* mRNA. H3K4me3 mark is not enriched at *Pparg* + 60 kb before differentiation but it becomes enriched by 48 h of differentiation (Supplementary Fig. 1f) which is coincident with the start of *Pparg2* transcription. These results indicate that *Pparg* + 60 kb region acts as an enhancer of *Pparg1* until 48 h of differentiation and then promotes expression of *Pparg2* after 48 h. At 48 h of differentiation, these H3K4me1 enriched regions received additional H3K27ac modifications (Supplementary Fig. 1d, tracks 4 and 6) indicating that these primed enhancers were converted into active enhancers. Time course ChIP-qPCR analysis confirmed that H3K27ac levels on *Cebpa* and *Pparg* enhancers were maintained very low during the early phase (~24 h) and were elevated at 48 h of differentiation (Supplementary Fig. 1g). Figure 1a illustrates three states of enhancers. Poised enhancers in embryonic stem cells (ESCs) are marked with H3K4me1 and H3K27me3. Enhancers transition from primed to active during early adipogenesis however the underlying mechanism has remained elusive.

**Identification of SETD5 as a negative regulator of adipogenesis.** To search for epigenetic enzymes that may keep enhancers in a primed state, we evaluated our prior transcriptome data for gene expression changes that occur during differentiation of 3T3-L1 adipocytes[21]. We found that mRNA expression of the putative histone modification enzyme SETD5 was down-regulated by two-fold by 48 h of differentiation in response to a strong MDI adipogenic cocktail which contains methylisobutyl-xanthine, dexamethasone, and insulin[14,35]. To investigate roles of SETD5 in adipogenesis, a V5-tagged SETD5 (SETD5-V5) retroviral expression construct (Fig. 1b) or an empty vector control were expressed using a weak LTR promoter[36] in 3T3-L1 preadipocytes. After stable transformation, we obtained two lines of SETD5-transduced 3T3-L1 preadipocytes that express 5.6-fold and 15-fold higher expression of *Setd5* mRNA compared to those in empty vector transduced preadipocytes (Supplementary Fig. 1h). Empty vector or SETD5-transduced 3T3-L1 preadipocytes were treated with the MDI mixture and subjected to Oil Red O (ORO) staining after 8 days of differentiation. This revealed that both high (15-fold) and moderate (5.6-fold) ectopic over-expression of SETD5 showed inhibitory effects on lipid accumulation (Supplementary Fig. 1i) accompanied with a blunting in the induction of *Cebpa* and *Pparg* (Supplementary Fig. 1j). Because the mRNA expression of endogenous *Setd5* is reduced during the early phase of differentiation[14], we reasoned that sustained expression of

SETD5 might inhibit adipogenesis. In the following experiments, we used the cell line expressing the moderate level of (5.6-fold) of SETD5-V5 unless otherwise stated. Immunoblot analysis and ORO staining show that the inhibitory effect of SETD5 on adipogenesis was only partially reversed by addition of the potent PPARγ synthetic full agonist troglitazone (Tro) (Fig. 1c, d). Retroviral expression of human PPARγ1 or PPARγ2 fully reversed the inhibitory effect of SETD5 on lipid accumulation (Supplementary Fig. 1k, l). Transcriptome analysis showed that induction of *Cebpb* and *Cebpd* genes, whose transcripts are induced at the very early phase of differentiation, were not affected by sustained SETD5 expression (Fig. 1e). By contrast, SETD5 prevented the late induction (48 h) of *Cebpa* and *Pparg*, as well as expression of PPARγ target genes such as *Cd36* (Fig. 1e, Supplementary Fig. 1m).

To complement the sufficiency results, we evaluated SETD5 necessity by predicting that a decrease in SETD5 expression would enhance differentiation of 3T3-L1 preadipocytes. Knockdown of SETD5 by two independent siRNAs resulted in prematurely early elevation of *Cebpa* and *Pparg* at 24 h of MDI induction at which time these genes remained repressed in control siRNA transfected preadipocytes (Fig. 1f, g). The proposed negative role for SETD5 in adipogenesis was also supported by a knockdown of SETD5 followed by induction with DEX alone, which is much less efficient at inducing differentiation than the complete cocktail[14,21,37,38]. The results are presented in Fig. 1h and show a marked increase in lipid accumulation relative to the control siRNA transfected preadipocytes that were treated with DEX only.

**SETD5 associates with NCoR–HDAC3 co-repressor complex and represses adipogenic genes.** The conserved SET domain binds the methyl-donating cofactor S-adenosyl methionine which is required for their histone methyltransferase activity. To determine whether the SET domain is required for SETD5-mediated inhibition of adipogenesis, we constructed a V5-tagged SET domain deletion mutant (ΔSET) SETD5 and analyzed its function in retrovirally transduced 3T3-L1 preadipocytes. Expression levels of WT and ΔSET mutant SETD5 proteins were very similar as confirmed by immunoblot analysis (Supplementary Fig. 1n). However, unexpectedly, ΔSET mutant blocked differentiation in response to the strong MDI cocktail very similarly to WT-SETD5 (Fig. 1i) and once again, this was not affected by inclusion of troglitazone (Tro) (Supplementary Fig. 1o). This result suggests that SETD5 inhibits adipocyte differentiation and lipid accumulation independently of its SET

domain. However, we found that deletion of amino acids 437–918 (Δ437–918) blunted the inhibitory effect of SETD5 (Fig. 1i, Supplementary Fig. 1o), indicating that this region is required for SETD5-mediated inhibition of lipid accumulation during adipocyte differentiation.

To begin to study the mechanism of adipocyte differentiation repression through this region of SETD5, we performed a comprehensive proteomics analysis to find possible proteins that may interact with SETD5 through this region. Extracts from preadipocytes expressing V5-tagged SETD5 wild-type (WT) or the two deletion mutants (ΔSET or Δ437–918) were immuno-precipitated with anti-V5 antibody and the profiles of interacting proteins were compared after mass spectrometry. This analysis showed that 509-512 proteins were identified as interacting proteins of WT, and the mutant forms of SETD5 (Supplementary Data 1). There were several peptide sequences that were immunoprecipitated in both WT and ΔSET SETD5 transduced preadipocytes but were absent in Δ437–918 transduced cells. These included peptides corresponding to NCoR (also known as NCoR1), SMRT (also known as NCoR2), TBL1 (also known as TBL1X), TBLR1, SKI, and HDAC3 (Fig. 1j, blue solid circle) which is known to interact with SETD5[27,30,39] suggesting that SETD5 associates with this co-repressor complex independent of its SET domain but dependent on the 437–918 region that was required for inhibition of adipogenesis. This is consistent with previous data showing a larger C-terminal deletion of SETD5 disrupts its interaction with HDAC3[27,30,39]. Co-immunoprecipitation analyses confirmed SETD5 interaction with NCoR2 and HDAC3 (Supplementary Fig. 1p). Components of polymerase-associated factor (PAF1) complex (CTR9, CDC73, PAF1, and LEO1)[40], which is known to interact with SETD5[27,39], were also found in SETD5 co-immunoprecipitates, although these were present in samples from WT and both mutants (Fig. 1j, solid black circles). The putative interaction between SETD5 with several other proteins (e.g. PRP4B, PLRG1) was SET domain dependent (Fig. 1j, solid orange circles) because these were not present in the ΔSET sample. Additionally, we observed the putative interaction with a multimeric ubiquitin E3 ligase APC/C complex, containing ANAPC1 and ANAPC2, and a ubiquitin E3 ligase RNF213 (Fig. 1j, solid pink circles, see also Fig. 3f) in samples from WT and both mutants.

Because the interaction with the NCoR and SMRT co-repressor complexes required the SETD5 domain which was essential to inhibit lipid accumulation, we explored this further. First, we depleted NCoR2 by siRNA in SETD5-transduced preadipocytes and showed that the suppression of lipid accumulation by SETD5 overexpression was reversed by NCoR2 depletion (Fig. 1k, l). Two other siRNAs targeted to Ncor2 also restored lipid accumulation in SETD5-transduced cells (Supplementary Fig. 1q, r), indicating that NCoR2 is involved in SETD5 mediated inhibition of adipogenesis possibly through the formation of protein complex with SETD5. These results indicate that SETD5 associates with NCoR and HDAC to inhibit Cebpa and Pparg expression and adipogenesis.

**SETD5 keeps enhancers in a primed state and its depletion leads to conversion into a hyperacetylated active state.** To interrogate mechanisms by which SETD5 might alter chromatin to inhibit Cebpa and Pparg expression, we performed ChIP-seq for H3K27ac before (0 h) and after differentiation (48 h) in two biological replicates of empty vector (control) and SETD5-transduced preadipocytes. Principal component (PC) analysis on ChIP-seq peaks showed that H3K27ac signals before differentiation are similar while the signals after differentiation induction are different between control and SETD5-transduced

preadipocytes (Supplementary Fig. 2a). Scatter plots confirmed reproducibility of each H3K27ac ChIP-seq sample in two biological replicates (Supplementary Fig. 2b). We identified 11,033 H3K27ac peaks that are elevated at 48 h of differentiation in control preadipocytes (Fig. 2a). These H3K27ac peaks were first classified into two groups based on the absence or presence of H3K4me3 mark, which represents active enhancers and promoters, respectively[5]. Among 9,546 active enhancers (i.e. H3K27 acetylated sites that lack H3K4me3 marks) at 48 h, forced expression of SETD5 repressed H3K27ac at 2,958 enhancers (31%) (Fig. 2a, class i) that includes enhancers of Cebpa and Pparg indicating that SETD5 inhibited H3K27 acetylation of enhancers during 48 h of early differentiation. SETD5, by contrast, increased H3K27 acetylation on certain enhancers (9%) (class iii). Among 1,487 promoters (i.e., the presence of both H3K27ac and H3K4me3), SETD5 repressed H3K27ac at 396 promoters (27%, class iv). These findings were further confirmed by statistical test of two biological replicates followed by the identification of differentially regulated H3K27ac regions using DeSeq2 (Supplementary Fig. 2c).

A gene ontology analysis showed that the enhancers whose H3K27ac levels were repressed by SETD5 (class i) were highly associated with lipid metabolic process, fat cell differentiation, and fatty acid metabolic process (Fig. 2b). Motif analysis showed that class i regions had motifs enriched in C/EBP and PPAR-RXR binding sequences (Fig. 2c), suggesting that SETD5-containing NCoR-HDAC3 repressor complex inhibits H3K27 hyperacetylation of enhancers (i.e. active state) bound by these transcription factors. Comparison of H3K27ac modification and genome wide binding sites of C/EBPβ and C/EBPδ (early adipogenic transcription factors) taken from the study of Mandrup and colleagues[19] showed that, among 2,958 H3K27ac sites that were found to be repressed in SETD5-transduced preadipocytes (i.e. sites kept in primed enhancers), 1,449 sites (49%) were bound by either C/EBPβ or C/EBPδ or both at 4 h of differentiation (Fig. 2d, left Venn diagram and right heatmap). Some of the SETD5-repressed H3K27ac sites were also binding sites for RXRα and PPARγ at 36 h of differentiation (Supplementary Fig. 2d, e). These results strongly suggest that SETD5 represses H3K27 acetylation to keep enhancers in a primed state during 24 h of differentiation through the binding sites of C/EBPβ and C/EBPδ (see also motif analysis and co-immunoprecipitation shown later in Fig. 4e, f). Active enhancers of Cebpa (e.g. +8 kb, +35 kb, and +53 kb) and Pparg (e.g. +60 kb, +158 kb) in control preadipocytes were repressed in SETD5-V5 transduced preadipocytes at 48 h (Fig. 2e, compare tracks 2 and 4, and Fig. 2f). Transcription factors C/EBPβ and C/EBPδ were recruited to some of the above enhancers (e.g. Cebpa + 8 kb and Pparg + 60 kb) in control cells as early as at 4 h of differentiation[19] (Fig. 2e, tracks 5 and 6, highlighted in yellow). SETD5 was also recruited to these C/EBPβ and C/EBPδ binding sites (shown later in Fig. 4c, d). To rule out the possibility that the decreased H3K27ac in SETD5 transduced preadipocytes is due to secondary effects from the defect of adipogenesis, we examined the effect of SETD5 on H3K27 acetylation of adipogenic enhancers in preadipocytes retrovirally expressing C/EBPα without inducing differentiation (Supplementary Fig. 2f). In control 3T3-L1 preadipocytes, viral expression of C/EBPα facilitated H3K27 acetylation on chromatin of Cebpa and Pparg enhancers (Supplementary Fig. 2g). In contrast, in SETD5 transduced 3T3-L1 preadipocytes, viral expression of C/EBPα did not induce H3K27 acetylation on these enhancers (Supplementary Fig. 2g). In a complementary experiment, siRNA mediated knockdown of SETD5 increased H3K27ac levels at enhancers of Cebpa (e.g. +8 kb and +53 kb) under DEX induction at 48 h induction (Fig. 2g). Together, these results indicate that SETD5 limits

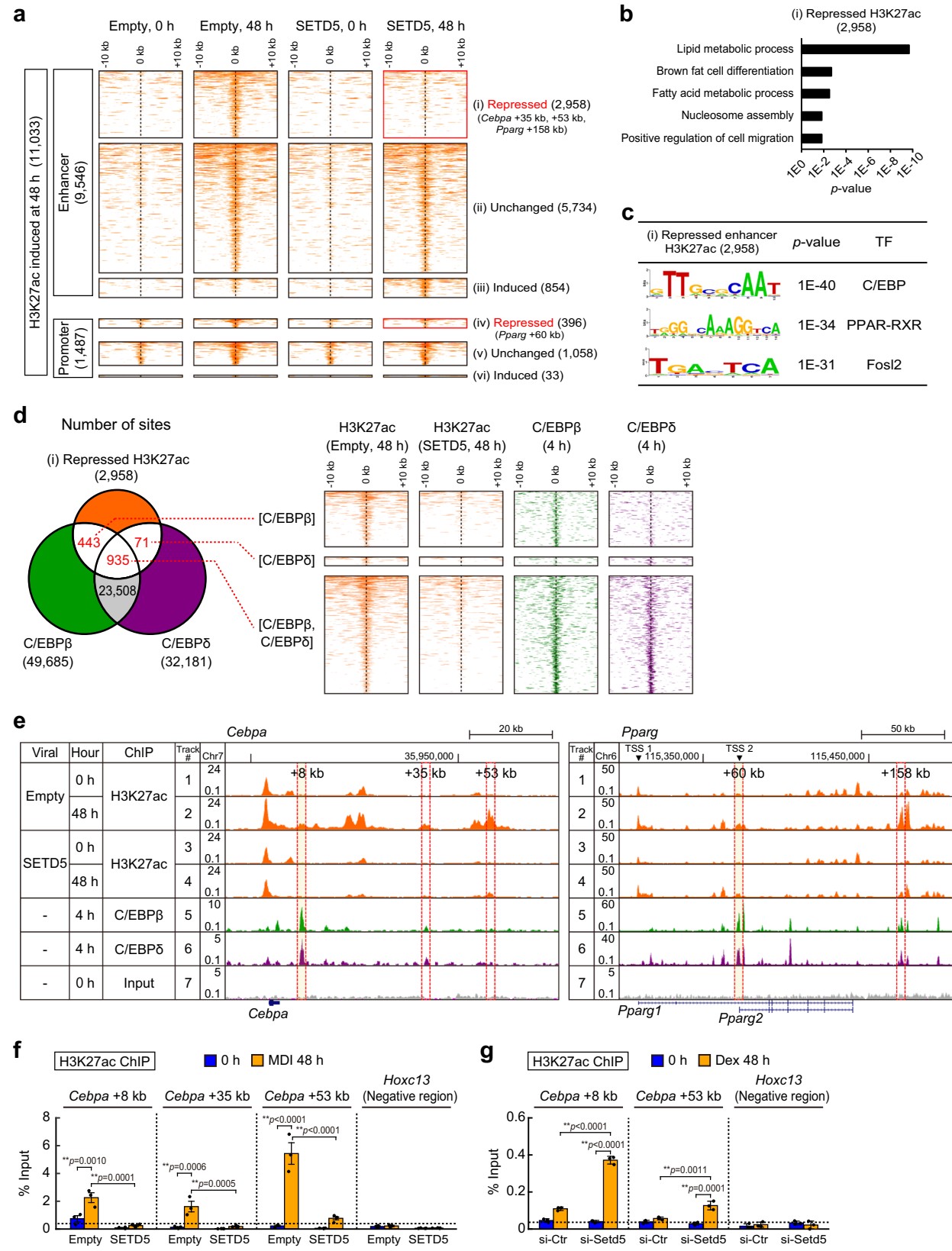

H3K27 acetylation at C/EBP binding site enhancers for *Cebpa* and *Pparg* genes.

**Transient stabilization and APC/C-mediated proteasomal degradation of SETD5 during early adipogenesis.** Because

protein levels of SETD5 determine either a primed or active enhancer state, a key issue is how SETD5 protein levels are controlled during adipogenesis. We noticed that compared to before MDI induction, endogenous SETD5 protein levels were transiently increased between 6 and 12 h and severely decreased at 48 h of differentiation (Fig. 3a). Curiously, we also noticed that

**Fig. 2 SETD5 inhibits H3K27 acetylation and enhancer activation of adipogenic genes. a** Heatmap representation of H3K27 acetylation during 3T3-L1 adipogenesis. Heatmap shows ChIP-seq data for H3K27ac at a 20-kb region centered on H3K27ac enriched region in 3T3-L1 preadipocytes transduced with empty virus or SETD5 at the indicated time after MDI induction. There were 11,033 H3K27ac enriched regions that were induced by differentiation stimuli (48 h) in control preadipocytes. These regions were classified into distal enhancer (9,546 regions) and promoter (1,487 regions) based on the absence and presence of H3K4me3 signals. H3K27ac enriched regions were further classified into repressed (<0.5-fold), induced (>2-fold), and unchanged by SETD5 transduction. **b, c** Gene ontology (GO) analysis (**b**) and HOMER motif analysis (**c**) of differentiation-induced and SETD5-repressed H3K27ac enriched enhancer regions. *p*-values of motif analysis were calculated as one-tailed. **d** Venn diagram and heatmap representation of differentiation-induced and H3K27ac enriched regions and C/EBPβ and C/EBPδ binding regions. 443 regions were bound by C/EBPβ, 71 regions were bound by C/EBPδ, and 935 regions were bound by both. Heatmap shows ChIP-seq data for H3K27ac, C/EBPβ, and C/EBPδ at a 20-kb region centered on H3K27ac enriched region. **e** Genome browser representation for H3K27ac, C/EBPβ, and C/EBPδ on adipogenic genes. *Pparg* gene has two transcription start sites (TSS). Tracks 1, 2, and 7 are the same data as Supplementary Fig. 1d. **f, g** ChIP-qPCR analysis of H3K27ac on *Cebpa* gene during adipogenesis. 3T3-L1 preadipocytes transduced with empty virus or SETD5 were subjected to ChIP-qPCR analysis of H3K27ac at the indicated time after MDI induction of differentiation (**f**). 3T3-L1 preadipocytes transfected with siRNA targeted to *Setd5* were subjected to ChIP-qPCR analysis of H3K27ac at the indicated time after Dex induction of differentiation (**g**). **d, e** ChIP-seq for C/EBPβ and C/EBPδ was from the previously published paper[19]. **f** Data are mean ± SEM (Empty *n* = 3; SETD5 *n* = 4; independent experiments). **g** Representative of two independent experiments. Data mean ± SD of three technical replicates. **f, g** One-way ANOVA with Tukey's multiple comparisons test. \*\**p* < 0.01. Source data are provided as a Source data file.

a similar pattern for ectopically expressed V5-tagged SETD5, suggesting that SETD5 protein stability may be differentially regulated during this short window of differentiation (Fig. 3b, compare lanes 3–5 and lanes 6–9). On the other hand, protein expression of NCoR2 and HDAC3, other subunits of the co-repressor complex, did not change during the early adipogenesis (Fig. 3b). Cell cycle analysis in Fig. 3c shows the first round of mitotic clonal expansion after MDI addition. $G_0$ arrested pre-adipocytes re-entered $G_1$ phase (0–12 h) followed by transition to S phase (12–21 h) and finally to $G_2/M$ phase (21–36 h). The timing of the transient increase of SETD5 protein (6–12 h) corresponds to the $G_1$ phase while its decline corresponds to S and $G_2/M$ phases (Fig. 3c, Supplementary Fig. 3a). Importantly, cell-cycling per se was not affected by ectopic SETD5 expression (Supplementary Fig. 3b).

We next asked whether this transient change in protein levels for SETD5 was due to changes in the rate of protein degradation because mRNA levels of endogenous SETD5 are reduced by only approximately 50% at these time points[14]. First, we measured SETD5 protein levels following MDI addition in the presence and absence of the proteasomal inhibitor MG132 which largely blocked the degradation of SETD5 protein (Fig. 3d, compare lanes 5–6 and 9–10) indicating that SETD5 is degraded via the ubiquitin-proteasome pathway. MG132 treatment increased a broad high molecular weight SETD5 signal recognized by anti-ubiquitin antibody, confirming that SETD5 is ubiquitinated and that proteasomal inhibition results in its accumulation (Fig. 3e).

Next, we searched for a possible ubiquitin ligase responsible for cell-cycle dependent SETD5 degradation. From our proteomic interaction data in Fig. 1j, two subunits of the anaphase-promoting complex/cyclosome (APC/C), ANAPC1 and ANAPC2 were found in immunoprecipitates of WT and the two mutant versions of SETD5 protein (i.e., wild-type, ΔSET, and Δ437–918) while ANAPC1 and ANAPC2 were not found in the immuno-precipitates from the control empty-virus transduced preadipocytes (Fig. 3f, Supplementary Fig. 3c). Among the APC/C subunits, the cullin subunit ANAPC2 and the RING domain subunit ANAPC11 compose the catalytic core and these two have critical roles in mediating the transfer of ubiquitin to substrates (Supplementary Fig. 3c)[22]. Another ubiquitin E3 ligase RNF213 was also found in immunoprecipitates of WT and two mutants of SETD5 protein (Fig. 3f).

mRNA levels of APC/C subunits themselves differed only slightly after MDI treatment and they were not altered by SETD5 overexpression (Supplementary Fig. 3d). However, The *Cdc20* coactivator of APC/C was very low during $G_1$ phase, and it was robustly increased during S and $G_2/M$ phase (Fig. 3g). The time

dependent induction of *Cdc20* mRNA (Fig. 3g) and protein (Fig. 3b) which is concomitant with the decrease of SETD5 protein levels suggested that degradation of SETD5 might be triggered by CDC20 induction and consequent APC/C activation. mRNA level of *Rnf213* was high before differentiation (0 h) and was decreased by 12 h of differentiation (Supplementary Fig. 3e) suggesting it is unlikely to mediate the degradation of SETD5 during S and $G_2/M$ phase. Co-immunoprecipitation analyses confirmed SETD5 interaction with ANAPC2 and ANAPC11 after differentiation induction (Supplementary Fig. 3f). Co-immunoprecipitation analysis also showed SETD5 interaction with APC/C coactivator CDC20 in the presence of MG132 (Supplementary Fig. 3g).

The stability of SETD5 protein was evaluated with a cycloheximide chase experiment where expression of the RING domain subunit ANAPC11 was knocked down and SETD5 levels were analyzed in the presence of cycloheximide (CHX), an inhibitor of translation. SETD5 was rapidly degraded in the presence of CHX in control siRNA transfected preadipocytes ($t_{1/2} = 2.0$ h), while SETD5 was markedly stabilized in ANAPC11 siRNA transfected preadipocytes ($t_{1/2} = 4.1$ h, Fig. 3h Supplementary Fig. 3h). Similarly, knockdown of APC/C coactivator CDC20 extended the half-life of SETD5 protein by a similar magnitude (Supplementary Fig. 3i, j). These results indicate that SETD5 is stabilized during $G_1$ phase where CDC20 level is low and APC/C is not active while during $G_2/M$ phase of the first mitosis of early adipogenesis, SETD5 levels fall reciprocally with the induction of CDC20 and activation of APC/C.

**SETD5 is transiently recruited to primed enhancers prior to enhancer activation.** The data so far suggests that SETD5 is recruited to primed enhancers early in adipogenesis to prevent their premature activation. To evaluate this directly, we analyzed genome-wide localization of SETD5 during early adipogenesis. For this purpose, we developed several monoclonal antibodies that recognize either the amino or carboxyl regions of SETD5 (Supplementary Fig. 4a, b). We chose one (IgG-F2104) that recognizes amino acids 43–93 and performed ChIP-seq for endogenous mouse SETD5 in preadipocytes at 6 h after MDI treatment when SETD5 levels are still high and we identified 3,179 putative SETD5 binding sites (Supplementary Fig. 4c, d, e). This represents a very low number relative to most other ChIP-seq genome-wide analysis so we performed ChIP-seq in V5-tagged SETD5 (SETD5-V5) retrovirally transduced preadipocytes at 0 h and at 6 h of induction. PC analysis and scatter plots showed reproducibility of each SETD5-V5 sample in two

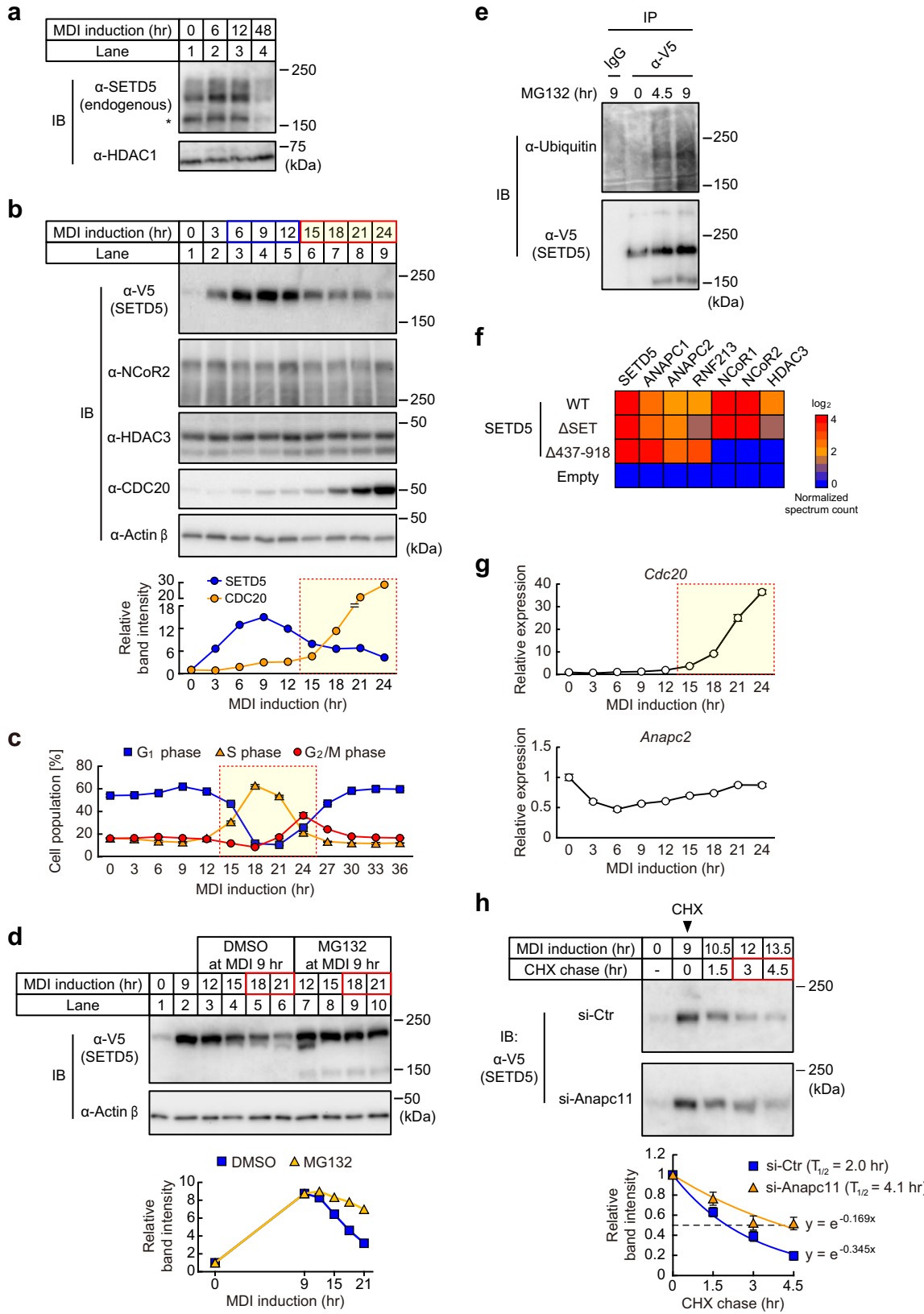

biological replicates (Supplementary Fig. 4f, g). Consistent with the elevated SETD5-V5 protein levels, the number of genome wide binding sites was increased from 15,245 to 32,043 at 6 h of induction (Fig. 4a).

Similar to endogenous SETD5 (Supplementary Fig. 4c), most of the SETD5-V5 binding sites were located at intron and intergenic regions (Fig. 4a). When we compared the SETD5-V5 binding sites (6 h) with the H3K27ac data of 11,033 differentiation-dependent H3K27 acetylated regions at 48 h post MDI, 6,586 regions (60%) were occupied by SETD5-V5 at 6 h of differentiation (Fig. 4b, left Venn diagram). SETD5 bound and unbound regions were further classified into three groups each based on the

**Fig. 3 SETD5 is transiently stabilized and degraded by APC/C and proteasome during the early adipogenesis. a** Immunoblot showing expression of endogenous SETD5 in 3T3-L1 preadipocytes. **b** Immunoblot showing expression of SETD5-V5, NCoR2, HDAC3, and CDC20 in SETD5-transduced 3T3-L1 preadipocytes. The graph shows the band intensity of SETD5-V5 and CDC20. **c** Cell cycle analysis of empty virus-transduced 3T3-L1 preadipocytes. The ratios of the cells at $G_1$, S, or $G_2$/M phase are shown. **d** Immunoblot showing expression of SETD5-V5 in SETD5-transduced 3T3-L1 preadipocytes treated with MDI and MG132. The graph shows band intensity of SETD5-V5. **e** Immunoblot showing ubiquitinated SETD5-V5. SETD5-transduced 3T3-L1 preadipocytes were treated with MDI and then MG132 at 9 h after MDI induction. Preadipocytes were subjected to immunoprecipitation using anti-V5 antibody and immunoblotting with anti-ubiquitin and anti-V5 antibodies. **f** Heatmap representing the abundance of APC/C subunits and RNF213 in SETD5-V5 immunoprecipitates. Spectrum count for each protein in proteomics analysis (Fig. 1j) was normalized by spectrum count for common region of SETD5 (a.a. 1 to a.a. 272 and a.a. 919 to a.a. 1441). **g** Expression of mRNAs of APC/C subunits during the early adipogenesis. mRNA expression of *Cdc20* and *Anapc2* in SETD5-V5-transduced preadipocytes were quantified by qPCR. **h** Immunoblot showing SETD5-V5 stability by the cycloheximide chase assay. 3T3-L1 preadipocytes transduced with SETD5-V5 were transfected with control siRNA or siRNA targeted to *Anapc11* and induced for differentiation with MDI mixture. Preadipocytes were treated with cycloheximide (CHX) at 9 h after MDI induction and subjected to immunoblot analysis with anti-V5. Representative of three independent experiments. The graph shows regression analysis of SETD5-V5 protein stability after CHX treatment. Data are mean ± SEM of three independent experiments. Protein half-life ($T_{1/2}$) was calculated based on exponential decay curve fit to the average data at each time point. **a, b, d** Equal loading of the protein was confirmed by blotting with anti-HDAC1 (**a**) or anti-actin β (**b, d**) antibody. **a, b, d, e, g** Representative of three (**a, b**) or two (**d, e, g**) independent experiments. **c, g** Data are mean ± SD of three technical replicates. Source data are provided as a Source data file.

effect of SETD5 on H3K27 acetylation (Fig. 4b, middle heatmap). SETD5 repressed H3K27 acetylation at 1,325 regions (class i) via direct binding (Fig. 4b, right pie chart). SETD5 repressed H3K27 acetylation at 2,029 regions (class iv) without direct binding, presumably via a mechanism involving chromatin looping. Importantly, at 6 h of differentiation, SETD5-V5 was recruited to H3K4me1 positive but H3K27ac negative regions (i.e. primed enhancers) and these regions were transformed to the dual marked H3K4me1 plus H3K27ac (i.e. active enhancers) at 48 h of differentiation where SETD5 protein was drastically reduced (Fig. 4b, middle heatmap). A genome browser snapshot of the ChIP-seq data also showed that SETD5-V5 was recruited to *Cebpa* + 8 kb and *Pparg* + 60 kb enhancers at 6 h of differentiation (Fig. 4c, tracks 1 and 2), the sites where C/EBPβ and C/EBPδ were also bound[19] (Fig. 4c, tracks 9 and 10). Consistent with total SETD5-V5 protein levels (Fig. 3b), recruitment of SETD5-V5 to these enhancers peaked at 6 h were decreased to very low levels at 24 h of differentiation (Fig. 4d). Motif analysis showed that SETD5-V5 binding sites were enriched in Jun-AP1, C/EBP, and TEAD binding sequences (Fig. 4e). Because C/EBPβ and C/EBPδ are two transcription factors induced highly by 6 h of differentiation (Supplementary Fig. 1a), we hypothesized that SETD5 may associate with primed enhancers through these proteins. Immunoprecipitation and immunoblot analysis using cross-linked nuclear lysates showed an interaction between C/EBPβ and SETD5 at 6 h of differentiation (Fig. 4f) and this interaction was diminished at 24 h of differentiation as SETD5 was degraded by this time. In addition, recruitment of SETD5-V5 to primed enhancers of *Cebpa* and *Pparg* at 6 h of differentiation were diminished via the knockdown of C/EBPβ and C/EBPδ (Fig. 4g and Supplementary Fig. 4h). The Δ437–918 SETD5 which lacks the domain required for the inhibition of adipogenesis (Fig. 1i) was similarly recruited to *Cebpa* and *Pparg* enhancers as wild-type SETD5 and the recruitment was increased at 6 h of differentiation compared to before differentiation (0 h) as demonstrated by V5 ChIP (Supplementary Fig. 4i). However, unlike wild-type SETD5 the Δ437–918 mutant could not inhibit the induction of *Cebpa* and *Pparg* mRNA expression and concomitant H3K27 acetylation at the chromatin of these genes (Supplementary Fig. 4j and 4k, respectively). These data indicate that SETD5 is recruited to enhancers via these transcription factors to prevent H3K27 hyperacetylation to keep the enhancers in their inactive but primed states.

**SETD5 is co-recruited to primed enhancers with NCoR-HDAC3 complex and diminishes prior to enhancer activation.** Our data so far suggest that SETD5 might function together

with the NCoR–HDAC3 complex to inhibit adipogenesis. Because its SET domain is not required, SETD5 must function through a mechanism that is independent of its methylation activity. Comparison of our SETD5-V5 ChIP-seq data with previously reported NCoR and HDAC3 ChIP-seq data in 3T3-L1 preadipocyte[32] revealed that approximately 60% of NCoR (11,002 of 18,384) and 71% of HDAC3 (4,069 of 5,765) genomic binding sites are biding sites for SETD5-V5 at 6 h of differentiation (Fig. 5a, left Venn diagram). Recruitment of NCoR, HDAC3, and SETD5 were all increased after 4–6 h of differentiation relative to those of 0 h (Fig. 5a, right heatmap). The ChIP-seq genome browser snapshot in Fig. 5b shows colocalization of SETD5-V5, NCoR, and HDAC3 to *Cebpa* and *Pparg* enhancers (e.g. *Cebpa* + 8 kb and *Pparg* + 60 kb). These data support that SETD5-NCoR-HDAC3 transiently co-localize on these enhancers during the priming step and prior to enhancer activation. Our ChIP-qPCR confirmed that the increased recruitment of NCoR2 and HDAC3 to *Cebpa* and *Pparg* enhancers at 6 h of differentiation consistent with recent ChIP-seq data[32] (Fig. 5c, d). Although SETD5-V5 binding was undetectable at 24 h of differentiation (Fig. 4d), NCoR2 and HDAC3 enhancer occupancy were still high at 24 h (Fig. 5c, d). Retrovirally expressed SETD5-V5 did not increase the recruitment of NCoR2 and HDAC3 to enhancers even at 6 h and 12 h of differentiation, where SETD5-V5 protein levels were the highest (Supplementary Fig. 5a, b). Biochemical analysis showed that SETD5 affected neither total nor HDAC3 specific histone deacetylase activities in vitro (Supplementary Fig. 5c, d). These data suggest that SETD5 may require NCoR-HDAC3 for its co-localization but SETD5 is likely not required for recruitment of NCoR-HDAC3 to enhancers.

To validate the contribution of APC/C mediated degradation of SETD5 to its recruitment to enhancers, ANAPC11 and CDC20 were depleted by siRNA mediated transfection. While SETD5 recruitment to enhancers at 24 h of differentiation was reduced compared with those at 6 h of differentiation (Figs. 4d, 5e, si-Ctr), knockdown of ANAPC11 or CDC20 kept SETD5 recruitment high at 24 h of differentiation (Fig. 5e) indicating that APC/C-mediated degradation of SETD5 contributes to its disappearance from primed enhancers prior to enhancer activation. These results indicate that SETD5-V5 increased co-localization with NCoR and HDAC3 at enhancers at 6–12 h of differentiation however, SETD5-V5 was lost through protein degradation by 24 h of differentiation (Fig. 5f).

**SETD5-containing NCoR-HDAC3 complex restricts HAT recruitment to primed enhancers.** Because these data suggested that SETD5 neither regulates NCoR-HDAC recruitment

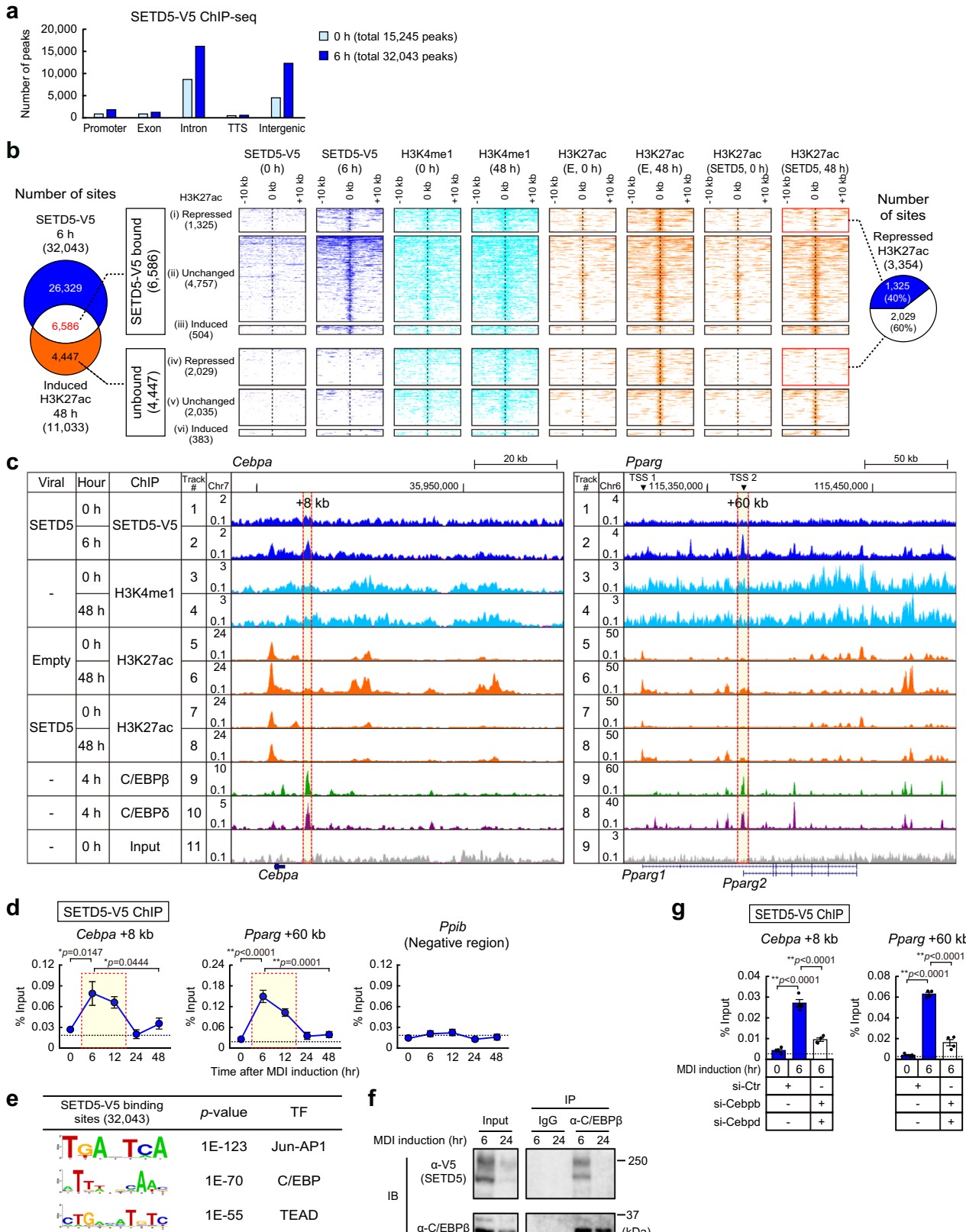

(Supplementary Fig. 5a, b) nor HDAC activity (Supplementary Fig. 5c, d) to decrease H3K27 acetylation on enhancer chromatin, we therefore hypothesized that SETD5 may limit recruitment of potential HATs to the enhancers. A comparison of ChIP-seq data for SETD5-V5 with the p300 HAT writer[32] showed that approximately 67% of p300 genomic binding regions (5,103 of

7,594 sites) were also binding sites for SETD5-V5 (Fig. 6a). A ChIP-seq genome browser snapshot shows both SETD5 and p300 HAT were recruited to primed enhancers (i.e. *Cebpa* + 8 kb and *Pparg* + 60 kb and + 158 kb) immediately after differentiation (4–6 h) (Fig. 6b). Unfortunately, the anti-p300 antibody used in the previous study[32] is currently unavailable, so we examined

**Fig. 4 SETD5 is recruited to primed enhancers and diminishes prior to enhancer activation. a** Genome-wide distribution of SETD5-V5 before (0 h) and 6 h of differentiation in 3T3-L1 preadipocytes. TTS, transcription termination site. **b** Venn diagram (*left*) and heatmap (*middle*) representations of genomic regions occupied by SETD5-V5, H3K4me1, and H3K27ac. Heatmap analysis showing ChIP-seq data for SETD5-V5, H3K4me1, and H3K27ac at a 20-kb region centered on SETD5-V5 binding regions. Pie chart (*right*) shows SETD5-V5 binding on H3K27 acetylated regions repressed by SETD5. **c** Genome browser representation for SETD5-V5, H3K4me1, H3K27ac, C/EBPβ, and C/EBPδ on *Cebpa* and *Pparg* genes in 3T3-L1 preadipocytes. Tracks 3–11 are the same data as Supplementary Fig. 1d or Fig. 2e. **d** ChIP-qPCR analysis of SETD5-V5 on *Cebpa* and *Pparg* genes during the early adipogenesis. 3T3-L1 preadipocytes transduced with SETD5-V5 were subjected to ChIP-qPCR analysis using anti-V5 antibody. Data are mean ± SEM (0, 6, 12, 48 h $n = 4$; 24 h $n = 3$; independent experiments). **e** Motif analysis of SETD5-V5 binding sites at 6 h of differentiation. Motif analysis was performed by Homer bioinformatics resources[3]. *p*-values were calculated as one-tailed. **f** Co-immunoprecipitation assay showing the interaction of SETD5-V5 with C/EBPβ. 3T3-L1 preadipocytes transduced with SETD5-V5 at the indicated time of differentiation were cross-linked with ethylene glycerol bis(succinimidyl succinate) and subjected to immunoprecipitation using anti-C/EBPβ antibody. Immunoprecipitates were decrosslinked and examined by immunoblotting using anti-V5 and anti-C/EBPβ antibodies. Representative of two independent experiments. **g** ChIP-qPCR analysis of SETD5-V5 on *Cebpa* and *Pparg* genes under knockdown of C/EBPβ and C/EBPδ. 3T3-L1 preadipocytes transduced with SETD5-V5 were transfected with control siRNA or siRNA targeting to *Cebpb* and *Cebpd* and subjected to ChIP-qPCR analysis using anti-V5 antibody. Data are mean ± SEM of four independent experiments. **d**, **g** One-way ANOVA with Tukey's multiple comparisons test. *$p < 0.05$; **$p < 0.01$. Source data are provided as a Source data file.

HAT recruitment of CBP which is highly related to p300[12]. A time course ChIP-qPCR analysis showed that while CBP recruitment to the primed enhancers for *Cebpa* was elevated by 6 h after differentiation in control preadipocytes (Empty) (Fig. 6c), however, H3K27ac levels were kept low until 48 h of differentiation (Supplementary Fig. 6a). Interestingly, SETD5-V5 expression limited both recruitment of CBP throughout differentiation and prevented the increase in H3K27ac at 48 h. This suggested that SETD5-V5 may restrict CBP recruitment to primed enhancers (Fig. 6c) and prevented the increase in H3K27ac (Supplementary Fig. 6a). Note that NCoR-HDAC3 co-repressor were also recruited as early as 6 h of differentiation but began to decrease after 24 h (Fig. 5c, d, f), while CBP recruitment remained elevated (Fig. 6c, Empty). We investigated whether Δ437–918 mutant SETD5 could inhibit CBP recruitment to the primed enhancers. The Δ437–918 mutant is recruited to primed enhancers similar to wild-type SETD5 (Supplementary Fig. 4i) but it does not interact with NCoR and HDAC3 (Fig. 1j) nor does it inhibit H3K27 acetylation on *Cebpa* and *Pparg* enhancers (Supplementary Fig. 4k). CBP HAT was similarly recruited to primed enhancers in Δ437–918 mutant-transduced preadipocytes to empty virus-transduced control preadipocytes (Fig. 6d), suggesting that SETD5 interaction with NCoR and HDAC3 restricts recruitment of CBP to primed enhancers (Supplementary Fig. 6b). To further complement the sufficiency results, we performed SETD5 knockdown by predicting that a decrease in SETD5 expression would elevate CBP recruitment to enhancers. SETD5 knockdown by siRNA elevated CBP recruitment to *Cebpa* and *Pparg* enhancers at 6 h and 12 h of differentiation (Fig. 6e), when SETD5 protein is stabilized (Fig. 3a). The elevated CBP recruitment by SETD5 knockdown was not observed at 24 h of differentiation (Fig. 6e), when APC/C and CDC20 mediates proteasomal degradation of SETD5 (Fig. 5f). SETD5 knockdown also resulted in prematurely early elevation of H3K27ac on *Cebpa* and *Pparg* enhancers at 12 h or 24 h of differentiation (Supplementary Fig. 6c) consistent with their mRNA expression (Fig. 1g). These results indicate that SETD5–NCoR–HDAC3 complex restricts accessibility of CBP, one of HATs, to primed enhancers, thereby regulating the time course of enhancer activation. When SETD5 protein diminishes via proteasomal degradation and consequently its recruitment to primed enhancers declines, CBP recruitment to primed enhancers increases to robustly drive master regulator gene expression and the adipogenic differentiation program (Fig. 6f).

**SETD5 inhibits adipogenesis in vivo.** To investigate the role of SETD5 in vivo, we followed the differentiation of 3T3-L1 pre-adipocytes into mature perilipin positive adipocytes following transplantation into nude mice (Fig. 7a). In this experiment, transplantation of control 3T3-L1 preadipocytes resulted in robust expression of perilipin (Fig. 7b, c). In contrast, 3T3-L1 preadipocytes transduced with the retroviral SETD5 expression vector prior to implantation to increase SETD5 expression levels resulted in a significant reduction in the appearance of perilipin positive adipocytes in vivo (Fig. 7c, d).

## Discussion

Strict temporal control of cell type-specific gene expression is essential for the exquisite timing and robust changes required for development and differentiation transitions and is mediated through the combinatorial activation and repression at key enhancers. HAT co-activators and HDAC co-repressors collaborate to ensure correct enhancer acetylation and gene activation. The response of preadipocytes to external differentiation stimuli controls the pattern of recruitment of HAT co-activators along with antagonistic HDAC co-repressors to fine-tune the differentiation response. Mechanisms to actively control the addition and removal of key histone acetylations provides an appropriately sensitive system to ensure accurate and precise timing for gene expression transitions that drive differentiation down an irreversible course[12,32]. In the current study, using protein interaction-based proteomics in the 3T3-L1 preadipocyte differentiation system, we identified SETD5 as a previously unappreciated component of the chromatin landscape that is required for adipocyte differentiation. Our studies show SETD5 interacts with the NCoR–HDAC3 complex to maintain enhancers for master genes of adipocyte differentiation in a histone hypoacetylated but "primed state" that is inactive. We show that the loss of SETD5 via proteasomal degradation synchronized with cell cycle triggers to influence enhancer hyperacetylation that is required for the transcriptional activation in lineage committed preadipocytes. SETD5 in an NCoR-HDAC3 complex restricts accessibility of CBP to enhancer chromatin but when SETD5 is lost from this complex via proteasomal degradation requiring the E3 ubiquitin ligase APC/C complex, associated or recruited HATs then hyperacetylate enhancers leading to subsequent gene activation. Importantly, the transition mechanism is regulated by the APC/C co-activator CDC20 which is induced in parallel with SETD5 degradation.

In the original model of signal-dependent gene activation, key signals drive a unidirectional switch between NCoR-HDAC co-repressor and HAT co-activator in a simple two-step process[41,42]. However, our study revealed that transition of adipogenic enhancers from a primed to active state is more dynamic and complex (Fig. 6f). Before differentiation (i.e. 0 h), NCoR-HDAC3 co-repressor and HAT co-activator are both recruited weakly to

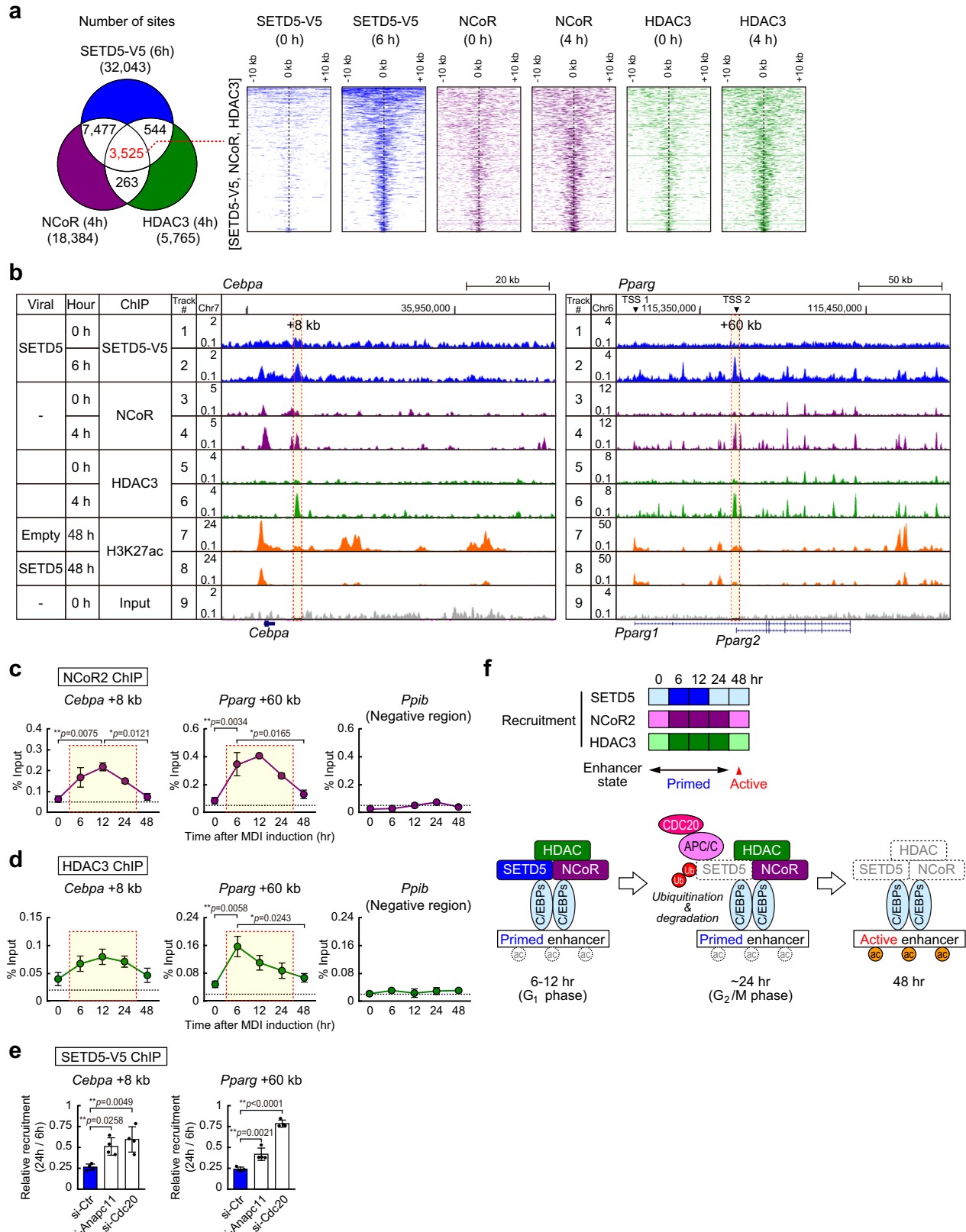

primed enhancers that are bound by pioneer transcription factors (i.e. C/EBPβ and C/EBPδ)[43] but are functionally silent. (i) In early differentiation (until 6–12 h), both NCoR-HDAC3-SETD5 co-repressor complex and CBP are tethered to primed enhancers where SETD5 restricts accessibility of CBP to enhancer chromatin and keeps histones hypoacetylated and held in a primed state. (ii)

As differentiation proceeds (i.e. after 24 h), as the first round of clonal expansion progresses, SETD5 is lost from the enhancer complex through active APC/C-mediated ubiquitination and degradation. (iii) In the absence of SETD5, the associated HAT leads to hyperacetylation and activation of enhancers and gene transcription (48 h). This mechanism allows precise temporal

**Fig. 5 SETD5 is co-recruited to primed enhancers with NCoR-HDAC3 complex and diminishes prior to enhancer activation. a** Venn diagram (*left*) and heatmap (*right*) representation of genomic regions occupied by SETD5-V5, NCoR, and HDAC3. Approximately 60% of NCoR and 71% of HDAC3 genomic binding regions were co-occupied by SETD5-V5. Heatmap analysis showing ChIP-seq data for SETD5-V5, NCoR, and HDAC3 at a 20-kb region centered on SETD5-V5 binding region. **b** Genome browser representation for SETD5-V5, NCoR, HDAC3, and H3K27ac on *Cebpa* and *Pparg* genes in 3T3-L1 preadipocytes. Tracks 1-2 and 7-9 are the same data as Fig. 4c. **c, d** ChIP-qPCR analyses of NCoR2 (**c**) and HDAC3 (**d**) on *Cebpa* and *Pparg* genes during the early adipogenesis. 3T3-L1 preadipocytes transduced with control empty virus were subjected to ChIP-qPCR analyses using anti-NCoR2 or anti-HDAC3 antibody. Data are mean ± SEM (0, 48 h $n = 5$; 6 h of NCoR2 $n = 4$; 6 h of HDAC3 $n = 5$; 12, 24 h $n = 3$; independent experiments). **e** ChIP-qPCR analysis of SETD5-V5 on *Cebpa* and *Pparg* genes under knockdown of ANAPC11 or CDC20. 3T3-L1 preadipocytes transduced with SETD5-V5 were transfected with control siRNA or siRNA targeting to *Anapc11* or *Cdc20* and subjected to ChIP-qPCR analysis at 6 h and 24 h of differentiation using anti-V5 antibody. ChIP signal was presented as relative SETD5 recruitment. Data are mean ± SD of four technical replicates. **f** Schematic model of SETD5-NCoR-HDAC3 complex formation and the loss of SETD5 on enhancers during the transition from primed to active states. Heatmap shows the degree of recruitment of SETD5, NCoR2, and HDAC3 to *Cebpa* and *Pparg* enhancers (based on Figs. 4d, 5c, and 5d). Darker color indicates more recruitment. **a, b** ChIP-seq for NCoR and HDAC3 was from the previously published paper[32]. **c–e** One-way ANOVA with Tukey's multiple comparisons test. *$p < 0.05$; **$p < 0.01$. Source data are provided as a Source data file.

control of adipogenic master regulator genes (i.e. *Cebpa* and *Pparg*) in synchronization with cell cycle regulation for cell fate decisions. Because SETD5 is ubiquitously expressed, this mechanism shown here in 3T3-L1 preadipocyte differentiation is likely to be applicable to other lineage committed cells. A SETD5-containing NCoR-HDAC3 complex may regulate enhancers recognized by other pioneer factors or cell type specific transcription factors in other lineage committed cells. This model can also explain why binding of pioneer factors do not facilitate transcription immediately until a critical time point of differentiation is reached, because colocalization of SETD5 with the NCoR-HDAC3 complex with C/EBPβ or C/EBPδ keeps enhancers in a primed state.

Histone modification enzymes can function as a scaffold for coactivator or repressor recruitment via enzymatic activity-independent mechanisms[44–47]. For example, the H3K9 demethylase JMJD1A acts as a cAMP-induced scaffold protein to stimulate enhancer-promoter chromatin looping of the *Adrb*1 gene in brown adipocytes through catecholamine mediated β-adrenergic receptor activation[46], and this bridging function is independently of the demethylase activity of JMJD1A. SETDB2 is predicted to be a H3K9 methyltransferase and it is induced during fasting in the liver through corticosteroid hormone-mediated glucocorticoid receptor activation and in turn mediates enhancer-promoter interaction of *Insig2a* gene, whose gene product inhibits SREBP activation[47]. Like JMJD1A or SETDB2, we demonstrate that SETD5 works independent of its putative methyltransferase activity to keep enhancers in a primed state. This is by restricting the accessibility of HATs to enhancer chromatin via the formation of a protein complex with NCoR and HDAC3. In response to the adipocyte differentiation cue, absolute protein expression level of SETD5 is strictly controlled by degradation through the ubiquitination by E3 ubiquitin ligase APC/C. When SETD5 is lost from the complex due to its degradation, HAT recruitment is accelerated and their writing activity dominates HDAC erasing activity leading to hyperacetylation and activation of enhancers (Figs. 5f and 6f). This suggests that the spatiotemporal dynamics of SETD5-NCoR-HDAC3 complex determines enhancer transition from a primed to active state.

Our studies document a previously unrecognized role for the APC/C complex and specifically CDC20 in coordinating the regulated degradation of SETD5 and enhancer hyperacetylation that occurs between 24 and 48 h after MDI treatment. Because SETD5 depletion leads to enhancer activation (Fig. 2g), we focused on the mechanism for SETD5 degradation during S and $G_2$/M phase. SETD5 is also induced during 3–12 h. after MDI treatment which is at a time when the mRNA levels for SETD5 decrease by about half. We do not know the mechanism for the early induction of SETD5 expression. A ubiquitin ligase RNF213 was also found in the co-immunoprecipitates with SETD5 in our proteomics analysis (Figs. 1j and 3f). In addition, *Rnf213* mRNA

levels rapidly declined during the first 12 h of differentiation (Supplementary Fig. 3e), therefore, RNF213 mediated SETD5 protein degradation might account for the early induction of SETD5 expression. Interestingly, RNF213 is an E3 ligase with a dynelin-like core and a distinct ubiquitin transfer mechanism and plays an important role in lipid metabolism modulating lipotoxicity, fat storage, and lipid droplet formation[48,49]. *RNF213* has been reported to be the major susceptibility factor for Moyamoya disease, a progressive cerebrovascular disorder that often leads to brain stroke in adults and children[50]. Whether RNF213 is responsible for the early induction of SETD5 protein levels remains to be determined.

While many of our conclusions for the genome wide studies rely on the SETD5-V5 data with the V5 antibody, we did observe the same trends with the IgG-F2104 antibody when analyzing endogenous SETD5. Thus, we are reasonably confident that the mechanism reflects the biological role for SETD5.

Several studies have noted that the cell cycle is linked to cell fate decisions via regulation of developmental transcription factors. For example, phosphorylation of Smad2/3 by cyclin and cyclin-dependent kinases renders embryonic stem cells susceptible to neuroectodermal differentiation[51]. Degradation of NeuroD2 by CDC20-APC/C regulates presynaptic differentiation[52]. We now showed that cell-cycle synchronized induction of CDC20 activates APC/C-mediated degradation of SETD5 for adipogenic cell fate decision. Thus, this represents a mechanism where the induction of CDC20, a coactivator of the APC/C protein degradation complex, connects the cell-cycle machinery to a critical cell fate decision via transcriptional and epigenetic mechanisms.

Upon environmental stress, cells reversibly regulate enhancer status to alter gene expression and adapt to key environmental cues. Under long-term cold stimuli, enhancers of thermogenic genes in adipocytes become active (both H3K4me1 and H3K27ac positive), while warming reverses the enhancer status back to a primed state (e.g. only H3K4me1 positive)[53] that can be rapidly re-activated upon repeated cold exposure. During reprogramming to induced pluripotent stem cells, fibroblast-specific active enhancers are returned to a poised or inactive state[54]. Our studies reveal a previously unknown role for regulated protein degradation of SETD5 in modulating enhancer transition during adipocyte differentiation. It will be important to address whether the SETD5-NCoR-HDAC3 complex regulation revealed here performs a similar function in other developmental and differentiation pathways and programs.

## Methods

A list of reagents and resource used is provided in Supplementary Table 1.

**Antibodies**. Mouse monoclonal IgG-F2104 and Z5721-234 against mouse SETD5 were developed by immunizing mouse with baculovirus particles displaying GP64

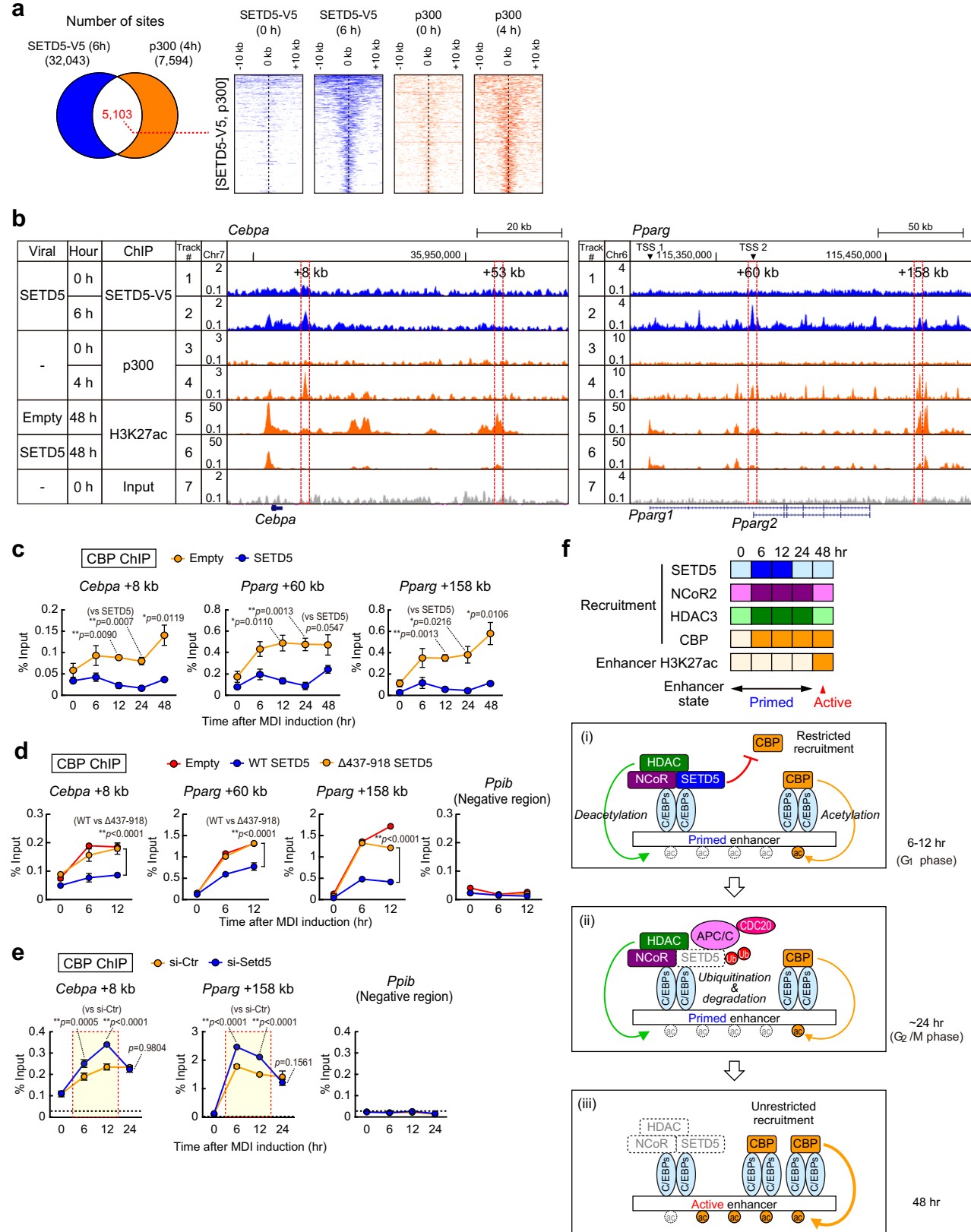

envelope fused to mouse SETD5 fragment (amino acids 49 to 93 or 1,241 to 1,291). IgG-F2104 which reacts with endogenous mSETD5 in immunoprecipitation and chromatin immunoprecipitation (ChIP) was used for ChIP-sequencing analysis.

**Cell culture and Oil Red O staining**. Mouse preadipocytes 3T3-L1, human HEK293, and Sf9 cell lines were purchased from the American Type Culture Collection (Manassas, VA, USA). Plat-E cell line was purchased from Cosmo Bio Co Ltd. 3T3-L1, HEK293, and Plat-E cells were maintained in Dulbecco's modified Eagle's medium (DMEM) (Sigma-Aldrich, St. Louis, MO, USA) supplemented with 10% fetal bovine serum (FBS) (Thermo Fisher Scientific, Waltham, MA) and penicillin/streptomycin (Nacalai Tesque, Kyoto, Japan) (basal medium) at 37 °C in a 5% $CO_2$ atmosphere in a humidified incubator. Sf9 cells were maintained in Grace's Insect Media (Thermo Fisher Scientific) supplemented with 10% FBS, 0.1%

**Fig. 6 SETD5–NCoR–HDAC3 complex restricts CBP recruitment to primed enhancers. a** Venn diagram (*left*) and heatmap (*right*) representation of genomic regions occupied by SETD5-V5 and p300. Approximately 64% of p300 genomic binding regions were co-occupied by SETD5-V5. Heatmap analysis showing ChIP-seq data for SETD5-V5 and p300 at a 20-kb region centered on SETD5-V5 binding region. **b** Genome browser representation for SETD5-V5, p300, and H3K27ac on *Cebpa* and *Pparg* genes in 3T3-L1 preadipocytes. Tracks 1-2 and 5-7 are the same data as Fig. 5b. **c, d, e** ChIP-qPCR analyses of CBP on *Cebpa* and *Pparg* genes during the early adipogenesis. 3T3-L1 preadipocytes transduced with control empty virus or SETD5 were subjected to ChIP-qPCR analyses using anti-CBP antibody **c**. 3T3-L1 preadipocytes transduced with control empty virus, wild-type SETD5, or Δ437-918 mutant were subjected to ChIP-qPCR analysis using anti-CBP antibody **d**. 3T3-L1 preadipocytes transfected with control siRNA or siRNA targeting to *Setd5* were subjected to ChIP-qPCR analysis using anti-CBP antibody **e**. **f** Model of enhancer transition from primed to active state during adipogenesis. Heatmap shows the degree of recruitment of SETD5, NCoR2, HDAC3, and CBP to *Cebpa* and *Pparg* enhancers and H3K27 acetylation (based on Figs. 4d, 5c, d, 6c, Supplementary Fig. 6a). Darker color indicates more recruitment and acetylation. (i) SETD5 forms a complex with NCoR-HDAC3 and keeps a hypoacetylation state by restricting the recruitment of CBP to primed enhancers. (ii) SETD5 in NCoR-HDAC complex on primed enhancers is ubiquitinated and degraded by APC/C. (iii) Degradation of SETD5 from NCoR-HDAC3 co-repressor complex allows the unrestricted recruitment of CBP and H3K27 acetylation and transit enhancers from primed to active state. **a, b** ChIP-seq data for p300 were from the previously published paper[32]. **c** Data are mean ± SEM (0, 6, 12 h $n = 3$: 24 h $n = 4$; 48 h $n = 5$; independent experiments). Unpaired two-tailed Student's *t*-test. **d, e** Representative of two independent experiments. Data are mean ± SD of three technical replicates. One-way ANOVA with Tukey's multiple comparisons test. *$p < 0.05$; **$p < 0.01$. Source data are provided as a Source data file.

Pluornic F-68 (Thermo Fisher Scientific), and penicillin/streptomycin in a 1-litter spinner flask at 27 °C. For adipocytes differentiation, 2 days after reaching confluence (day 0), 3T3-L1 preadipocytes were treated with differentiation medium containing insulin (1 µg/ml), 0.25 µM dexamethasone (DEX), and 0.5 mM iso-butylmethylxanthine (MDI mixture) for 48 h followed by treatment with insulin (1 µg/ml) alone with medium replacement every 2 days as described previously[38,37]. On day 8 of differentiation, the cells were stained with Oil Red O (ORO) as described previously[37,38]. To analyze protein stability, 3T3-L1 preadipocytes were treated with MG132 (10 µM) or CHX (10 µg/ml).

**Immunoblot analysis**. Whole cell lysate (WCL) or nuclear fraction was prepared from cells at indicated day of differentiation[21,37]. Aliquots of WCL or nuclear fraction were subjected to immunoblot analysis as described. Primary antibodies used: anti-SETD5 mouse monoclonal antibody (mAb) IgG-F2104 (1 µg/ml), anti-SETD5 mouse mAb IgG-Z5721-234 (1 µg/ml), anti-TBP mouse mAb (Novus Biologicals, NB500-700, clone 1TBP18, 5 µg/ml), anti-V5 mouse mAb (Thermo Fisher Scientific, R960-25, 1 µg/ml), anti-Histone H3 rabbit polyclonal antibody (pAb) (Abcam, ab1791, 0.05 µg/ml), anti-HDAC1 mouse mAb (Santa Cruz Biotechnology, sc-8410, clone H-11, 0.4 µg/ml), anti-NCoR2 rabbit pAb (Abcam, ab5802, 1 µg/ml), anti-C/EBPβ mouse mAb (Santa Cruz Biotechnologies, sc-7962, clone H-7, 0.4 µg/ml), anti-multiple ubiquitin mouse mAb (Medical and Biological Laboratories, D058-3, clone FK2, 1 µg/ml), anti-ANAPC2 rabbit pAb (Cell Signaling Technology, 12301, 1:500), anti-ANAPC11 rabbit mAb (Cell Signaling Technology, 14090, clone D1EQ7, 1:500), anti-CDC20 rabbit mAb (Cell Signaling Technology, 14866, clone D6C2Q, 1:1000), anti-Actin β mouse mAb (Sigma-Aldrich, xx, clone AC15, 1:5000), and anti-FLAG mouse mAb (Sigma-Aldrich, F3165, clone M2, 1 µg/ml). Secondary antibodies used: anti-mouse IgG-horse radish peroxidase (HRP) conjugate (Sigma-Aldrich, A4416, 1:20,000) and anti-rabbit IgG-HRP conjugate (Sigma Aldrich, A0545, 1:20,000). Immunoblots were visualized by chemiluminescence using Super Signal West Dura Extended Duration Substrate (Thermo Fisher Scientific), and luminescent images were analyzed by ImageQuant LAS 4000mini (GE Healthcare, Chicago, IL, USA). Equal loading of the proteins was confirmed by the detection of histone H3, TBP, or HDAC1. For quantification of immunoblot, ImageJ1.53k software was used.

**RNA interference**. The duplexes of each small interfering RNA (siRNA), targeting mouse *Setd5* mRNA (corresponding to nucleotide 2,061–2,085 and 3,859–3,883 from start codon; Thermo Fisher Scientific), *Ncor2* (MSS209220), *Cebpb* (corresponding to nucleotide 1,179–1,203 from start codon), *Cebpd* (MSS273628), *Anapc11* (MSS287826), or *Cdc20* (MSS235609) and control siRNA (Med GC Duplexes #2 12935-112) were transfected into cells using Lipofectamine RNAi MAX reagent (Thermo Fisher Scientific) as described[37,38]. ON-TARGETplus siRNA targeting mouse *Ncor2* (J-045364-05, J-045364-06) was purchased from Dharmacon (Lafayette, CO, USA).

**Plasmid construction and retroviral transduction**. To construct retroviral expression vector for SETD5 containing V5-tag at COOH-terminus and FLAG-tag an NH$_2$-terminus, mouse SETD5 coding sequence was cloned to pMXs-IRES-puro and pMXs-IRES-zeo vectors driven weak LTR promoter[36,46], respectively. Deletion mutants were constructed by inverse PCR using KOD Plus neo DNA polymerase (Toyobo, Osaka, Japan). To construct retroviral expression vector for C/EBPα, PPARγ1, and PPARγ2, coding sequences of mouse C/EBPα, human PPARγ1, and human PPARγ2 were cloned to pMXs-puro[36]. To construct transient expression plasmids for SETD5 containing the FLAG-tag at NH$_2$-terminus and HDAC3 containing V5-tag at COOH-terminus, mouse SETD5 coding sequence and mouse HDAC3 coding sequence were cloned to pCAG-IRES-Bsd and pcDNA3.1,

respectively. All PCR-generated constructs were verified by DNA sequencing. Retroviruses were produced in Plat-E cells[55]. 3T3-L1 preadipocytes were infected by retrovirus and selected with puromycin or zeocin as described[37,38].

**Identification of SETD5-interacting proteins**. Retrovirally transduced 3T3-L1 preadipocytes were grown in a 15-cm dish and harvested. The pellet of cells was washed once with PBS and frozen at −80 °C until use. All subsequent operations were carried out on ice or at 4 °C. Each cell pellet was thawed out and allowed to swell in hypotonic buffer B (10 mM HEPES (pH 7.9), 10 mM KCl, 1.5 mM MgCl$_2$, 1 mM EDTA, 1 mM EGTA, 1 mM dithiothreitol, and a mixture of protease inhibitors (cOmplete, Mini, EDTA-free; Merck, Darmstadt, Germany)) for 30 min, passed through a 25-gauge needle five times, and centrifuged at 20,000 g for 1 min. The pellet was resuspended in 0.2 ml of buffer C (10 mM HEPES (pH 7.9), 10 mM KCl, 1 mM MgCl$_2$, 0.5% (v/v) Nonidet P-40, and a mixture of protease inhibitors) and sonicated on ice 10 times using 10-s pulses using a Sonifier cell disruptor model 250 (Branson Ultrasonics, Danbury, CT, USA), and then the debris was removed by centrifugation. Supernatants were collected, and buffer was exchanged to 50 mM Tris-HCl (pH 8.0), 100 mM KCl, 0.1 mM EDTA, 5% glycerol, 0.1% Nonidet P-40 by Econo-Pac 10DG (Bio-Rad, Hercules, CA, USA), ultrafiltrated by Amicon Ultra-4 MWCO 30 K (Merck Millipore, Burlington, MA, USA), and used for immunoprecipitation. The samples were incubated with control IgG or anti-V5 epitope antibody (Thermo Fisher Scientific, R960-25, 2 µg) cross-linked with Dynabeads protein G (Thermo Fisher Scientific) and rotated for 18 h at 4 °C. The beads were washed three times with buffer containing 20 mM HEPES (pH 7.9), 200 mM KCl, 2 mM EDTA, 0.1% Nonidet P-40, after which the protein complexes were eluted with buffer containing 62.5 mM Tris-HCl (pH 6.8), 2% SDS, and 25% glycerol for 5 min at 95 °C. Each eluate was precipitated with methanol and chloroform, washed with ice-cold acetone, and centrifuged at $2,000 \times g$ for 10 min as described[56]. Each pellet was airdried and resuspended in 25 mM NH$_4$HCO$_3$ buffer containing 25% (v/v) CH$_3$CN at room temperature. The samples were reduced in 1.2 mM tris(2-carboxyethyl)phosphine for 15 min at 50 °C and alkylated in 3 mM iodoacetamide for 30 min at room temperature, respectively. The samples were digested overnight with 100 ng of trypsin (Promega, Madison, WI, USA) at 37 °C. Aliquots of trypsinized samples were analyzed by liquid chromatography-tandem mass spectrometry (LC/MS/MS) as we described previously[57]. Protein quantification was done by processing the acquired LC-MS data with the software Progenesis LC-MS (version 2.6; Nonlinear Dynamics, Newcastle, UK). MS/MSspectra from features with charge +2, +3, and +4 were exported and searched against the Swiss-Prot database using Mascot (version 2.3; Matrix Science).

**Co-immunoprecipitation**. Whole-cell lysate (WCL) or nuclear fraction was prepared from 3T3-L1 preadipocytes transduced with SETD5-V5 before differentiation induction. For detection of SETD5 interaction with CDC20, preadipocytes were induced with MDI for 9 h and then treated with 10 µM MG132 for 12 h before harvesting. WCL or nuclear fraction was subjected to immunoprecipitation using anti-V5 antibody (Thermo Fisher Scientific, R960-25, 2 µg) or control mouse IgG. Immunoprecipitates were eluted with SDS sample buffer and subjected to immunoblot analysis using indicated antibodies. For detection of SETD5 interaction with C/EBPβ, 3T3-L1 preadipocytes transduced with SETD5-V5 were cross-linked with 1.5 mM ethylene glycol bis(succinimidylsuccinate) (Thermo Fisher Scientific) for 30 min at followed by second cross-linking by addition of 1% formaldehyde for 10 min. After cross-linking, nuclear fraction was prepared and subjected to immunoprecipitation with Dynabeads Protein G (Thermo Fisher Scientific) conjugated with anti-C/EBPβ (Santa Cruz Biotechnologies, sc-150×, 6 µg) or control rabbit IgG. Immunoprecipitates were eluted with SDS sample buffer, de-crosslinked at 37 °C for 3 h, and subjected to immunoblot analysis.

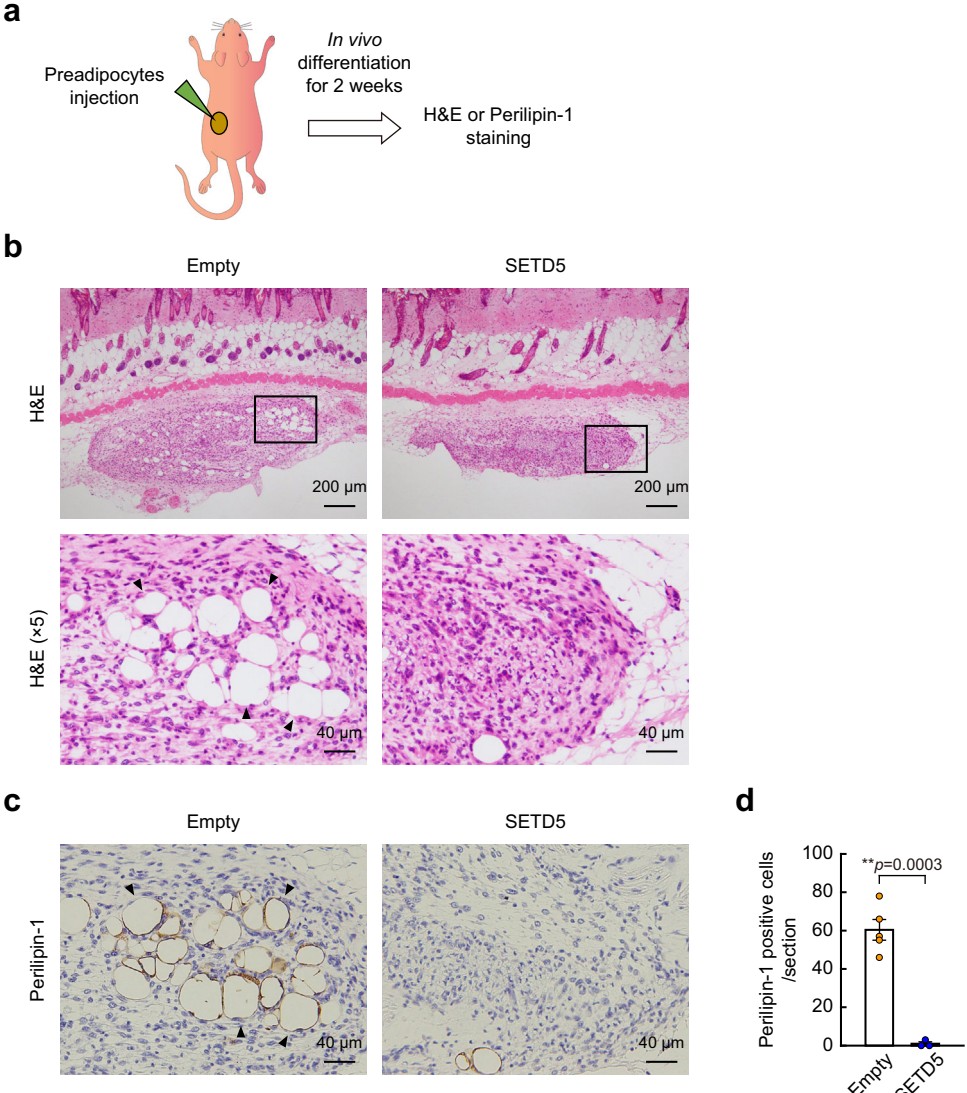

**Fig. 7 SETD5 inhibits adipogenesis in vivo. a** Transplantation of 3T3-L1 preadipocytes into nude mice. 3T3-L1 preadipocytes transduced with control empty virus or SETD5-V5 were cultured for two days with MDI cocktail. Subsequently, harvested cells were implanted with HydroMatrix and injected into the subcutaneous region on the upper or lower back of nude mice. **b, c** Two weeks after transplantation, mice were dissected, and isolated transplanted fats were subjected to haematoxylin and eosin (H&E) staining (**b**) and immunohistochemistry using anti-perilipin-1 antibody (**c**). Representative images of transplants from the lower back are shown. **d** Quantitation of perilipin-1 positive adipocytes. Data are mean ± SE in the section of transplants from empty virus transduced ($n = 5$) or SETD5 transduced ($n = 3$) preadipocytes Unpaired two-tailed Student's $t$-test. **$p < 0.01$. Source data are provided as a Source data file.

**HDAC activity assay**. For total HDAC activity, nuclear fraction prepared from 3T3-L1 preadipocytes transduced with control empty virus or SETD5-V5 was subjected to in vitro assay using HDAC fluorometric activity assay kit (Cayman Chemical, Ann Arbor, MI, USA). For HDAC3 activity, HEK293 cells were co-transfected with plasmids encoding FLAG-SETD5 and HDAC3-V5, and nuclear fraction was subjected to immunoprecipitation with anti-V5 antibody (Thermo Fisher Scientific, R960-25, 5 μg) or control mouse IgG. Co-immunoprecipitates of HDAC3-V5 were subjected to in vitro assay using HDAC fluorometric activity assay kit. Background signal of mouse IgG control was subtracted from fluorescent signal of HDAC3-V5 co-immunoprecipitate to obtain HDAC3 activity.

**Chromatin immunoprecipitation**. The chromatin immunoprecipitation (ChIP) assays were performed as described previously[38,46,37]. For ChIP using anti-H3K27ac, anti-H3K4me1, and anti-H3K4me3 antibodies, cells were cross-linked with 0.5% formaldehyde, while for ChIP using anti-CBP antibody, cells were cross-linked with 1% formaldehyde for 10 min. For ChIP using anti-V5-tag, anti-SETD5, anti-NCoR2, and anti-HDAC3 antibodies, cells were cross-linked with 1.5 mM ethylene glycol bis(succinimidylsuccinate) (Thermo Fisher Scientific) for 30 min at followed by second cross-linking by addition of 1% formaldehyde for 10 min. After cross-linking, nuclear fraction was prepared and sheared to 200-300 bp (for anti-

H3K27ac and anti-H3K4me3) and ~2 kb (for other antibodies) by using SONIFIER 250 (Branson). Sonicated nuclear fraction was incubated overnight with each antibody pre-bound to Dynabeads Protein G (Thermo Fisher Scientific). Anti-bodies used: anti-H3K27ac mouse mAb 9E2H9[58] (5 μg of antibody/10 μg of DNA), anti-V5 mouse mAb (Thermo Scientific, R960-25, 5 μg of antibody/25 μg of DNA), anti-SETD5 mouse mAb IgG-F2104 (120 μg of antibody/40 μg of DNA), anti-H3K4me1 rabbit pAb (Abcam, ab8895, 2 μg of antibody/25 μg of DNA), anti-H3K4me3 rabbit pAb (Merck Millipore, 07-473, 3 μl of antibody/30 μg DNA), anti-NCoR2 rabbit pAb (Abcam, ab5802, 3 μg of antibody/25 μg of DNA), anti-HDAC3 rabbit pAb (Abcam, ab7030, 25 μg of antibody/12.5 μg of DNA), and anti-CBP rabbit mAb (Cell Signaling Technologies, 7425, clone D9B6, 4 μl of antibody/15 μg of DNA). ChIP DNA was purified using QiAquick PCR purification kit (Qiagen), and the concentration was measured by Qubit double-stranded DNA high sensitivity assay kit (Thermo Fisher Scientific).

**qPCR, ChIP-qPCR, and ChIP-seq analyses**. qPCR was carried out in 384-well plates using ABI PRISM 7900HT sequence detection system (Applied Biosystems). All reactions were done in triplicate. mRNA expression was presented as fold change relative to indicated control after normalization to cyclophilin[46]. ChIP signals were presented as input percent as described[37]. All primer sequences used

in this article are listed in Supplementary Tables 2 and 3. ChIP-seq library was prepared using TruSeq ChIP library preparation kit (Illumina) or KAPA Hyper Prep Kit (Kapa Biosystems) according to the manufacturer's instructions. For ChIP-seq using anti-V5-tag and anti-SETD5 antibodies, ChIP DNA was sheared to ~200 bp by Covaris Acoustic Solubilizer (Covaris) before library preparation. ChIP-seq was performed on Illumina Genome Analyzer IIx using Sequencing Control Software v2.10.17 or HiSeq 2500 using HiSeq Control Software v2.2.58 as previously described[46,37].

**Microarray analysis.** Transcriptome analysis was performed using the Affymetrix GeneChip system according to the manufacturer's instructions as described[46]. Labeled cRNA probes were hybridized to Mouse Genome 430 2.0 array (Affymetrix) and scanned by GeneChIP scanner 3000 (Affymetrix). To calculate the average difference for each gene probe, GeneChip Analysis Suite software version 5.0 (Affymetrix) was used.

**Computational data analysis.** For ChIP-seq data processing, all bound DNA fragments were mapped to UCSC build mm9 (NCBI Build37) assembly of the mouse genome by the mapping program ELAND in CASSAVA pipeline 1.8.2 (Illumina) or Bowtie2.3.4.3[59] based on the 5′-side 36-bp sequences. Trimmomatic0.39 was used for trimming the reads[60]. Samtools1.9 was used to produce sorted bam files of the mapped reads[61]. ChIP-seq signals were plotted in reads per million mapped reads (RPM) and displayed on UCSC genome browser. To identify ChIP-seq enriched regions of RXRα and PPARγ, MACS1.4.2 program, a method appropriate for sharp peaks, was used under default setting[62,63]. For H3K27ac, SETD5-V5 and SETD5, SICER1.1 or SICER2-1.0.2 program, a method appropriate for broad peaks[64,65], was used under the following parameters: window size 200 bp; gap size 400 bp (H3K27ac) and 600 bp (SETD5-V5 and SETD5); E-value threshold, 100. Differentially regulated regions were identified by using DESeq2.1.30.1 program with an adjusted p-value < 0.05[66]. Previously reported ChIP-seq data for H3K27ac, C/EBPβ, C/EBPδ, HDAC3, NCoR, and p300[19,32,67] were used to analyze genomic binding regions by using MACS2.2.7.1 program[62,63]. Each enriched region was annotated to proximal genes within 50 kb. Bedtools2.27.1 was used to find overlapping peaks[68]. For a gene ontology analysis, DAVID 6.7 bioinformatics resources were used[69], and the p-value is corrected for multiple hypothesis testing using the Benjamini-Hochberg method. Motif analysis was performed by using HOMER4.10.4 program[3].

**Flow cytometric analysis.** After MDI induction, the 3T3-L1 cells were trypsinized for 5 min at 37 °C. Harvested cells were washed by PBS and fixed with 70% ethanol at 4 °C overnight. The cells were then wash by PBS three times and stained by Cell Cycle Assay Solution Blue (Dojindo) at 37 °C for 15 min. Cells were analyzed using LSRFortessa (BD Bioscience) equipped with 405 nm laser and 450/50 nm filter. Data were analyzed using FlowJo v10 software (BD Bioscience).

**Fat transplantation.** All animal studies were approved by the Animal Care and Use Committee of Tohoku University. Male nude mice (BALB/cSlc-nu/nu, purchased from Japan SLC, Inc) were fed standard chow (CE-2, CLEA Japan Inc.) ad libitum in a temperature- and humidity-controlled environment with a 12 h light/12 h dark cycle (08:00–20:00) at constant temperature (23 °C). Transplantation of cultured cells was operated as previously reported[70]. 3T3-L1 preadipocytes transduced with empty virus or SETD5 were cultured in differentiation medium for two days. Subsequently, cells were collected by scraping and resuspended in the basal medium containing 1 μg/ml insulin and mixed with 0.25% HydroMatrix Peptide Cell Culture Scaffold (Sigma-Aldrich). Cells (1.5–2 × 10^6 cells) were injected into the subcutaneous region on the upper or lower back of nude mice at 6 weeks of age. Two weeks after injection, transplanted cells were isolated with skin and subjected to histological analysis.

**Histology and immunohistochemistry.** Isolated tissues were fixed in 10% formalin overnight at 4 °C. After dehydration, tissues were embedded in paraffin and sliced to a thickness of 3 μm. Sliced sections were stained with hematoxylin and eosin (H&E) following the standard protocol. For immunohistochemistry, slides were deparaffinized, rehydrated and incubated with anti-Perilipin-1 (Cell Signaling Technologies, 9349, clone D1D8, 1:300 dilution) overnight at 4 °C. Endogenous peroxidase activity was quenched with 0.3% $H_2O_2$ and methanol at the room temperature for 20 min. Histofine MAX-PO (R) (Nichirei Bioscience Inc.) and diaminobenzidine tetrahydrochloride (DAB) were used for visualization of the binding of the perilipin antibody. Counter-staining was performed with hematoxylin. Photos of H&E staining and immunostaining were captured with BX51 (OLYMPUS) and cellSens standard 1.17 (OLYMPUS).

**Quantification and statistical analysis.** Data are shown as mean ± SEM for independent experiments or mean ± SD for technical replicates. Statistical test was performed by unpaired two-tailed Student's t-test or one-way ANOVA with Tukey's multiple comparisons test. p-values denoted as *p < 0.05; **p < 0.01.

**Reporting summary.** Further information on research design is available in the Nature Research Reporting Summary linked to this article.

## Data availability

The data that support this study are available from the corresponding authors upon reasonable request. Gene expression microarray data and ChIP-seq data for H3K27ac, SETD5-V5, H3K4me1, and SETD5 were deposited in the Gene Expression Omnibus (GEO) database with accession numbers GSE183849. Other ChIP-seq data were already published and deposited in the GEO database GSE73434[37] or the DNA data bank of Japan (DRA000378)[71]. Previously reported ChIP-seq data for H3K27ac, C/EBPβ, C/EBPδ, HDAC3, NCoR, and p300 are available in the GEO database GSE27826, GSE95533, and GSE56872[19,32,67]. Proteomics data were deposited in ProteomeXchange with the accession number PXD029279. Source data are provided with this paper.

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

## Acknowledgements

We thank Reiko Kuwahara, Akashi Taguchi-Izumi, and Choi Hyunmi for technical assistance; Minori Yoshio for secretary assistance; Dr. Toshio Kitamura for pMXs-puro plasmid; and Dr. Kazuhisa Takeda and other members of the Sakai laboratory for helpful discussion. We thank Dr. Kai Ge for the helpful discussion; Dr. Yasuo Oguri for advisement on the transplantation experiment. The super-computing resource was provided by Human Genome Center (the University of Tokyo). We are grateful to the Biomedical Research Core of Tohoku University Graduate School of Medicine for supporting the immunohistochemical experiments. This study was supported by grants-in-aid for scientific research (to Y.M. and J.S.), and for scientific research on innovative areas (to J.S.) from the Ministry of Education, Science, Sports and Culture (MEXT), AMED-CREST, under Grant Number JP20gm1310007 (to Y.M., S.T., T.Y., and J.S.), Japan and grants from the Naito Foundation and Takeda Science Foundation.

## Author contributions

Y.M. and J.S. directed the study and wrote the paper. T.F.O. critically commented and edited the paper. Y.M., R.I., A.Y., R.Y., K.M., Y.A., A.U., H.H., J.Z., M.A., G.Y., H.T., H.F., and S.O. performed experiments. T.T. and T.Kawamura performed proteomics analysis. H.A. and Y.W. supported ChIP-seq, and Y.M., R.N., S.Y., S.T., and C.Y. analyzed ChIP-seq data. H.K. provided materials. T.Y., S.I., T.Kodama, T.I., and K.N. commented on the paper. All authors reviewed the results and approved the final version of the manuscript.

## Competing interests

The authors declare no competing interests.
