## [Peer Review File · Nature Communications]

REVIEWER COMMENTS

Reviewer #1 (Remarks to the Author):

During cellular differentiation, enhancers are modified by the addition/subtraction of histone marks that determine whether they are poised, primed, or active. A very well studied cell differentiation program is the conversion of preadipocytes into adipocytes. In these studies, Matsumura et al. find that the complex formed between SET domain-containing 5 (SETD5) and the NCoR-HDAC3 co-repressor prevents histone acetylation of enhancers for two master adipogenic regulatory genes *Cebpa* and *Pparg* early during adipogenesis. Loss of SETD5 from the complex caused enhancer hyperacetylation, and SETD5 protein levels were transiently increased and rapidly degraded prior to enhancer activation. Induction of the CDC20 co-activator of ubiquitin ligase leads to APC/C mediated degradation of SETD5 during the differentiation transition, which may provide a molecular switch that facilitates adipogenesis.

These studies are of interest for at least two reasons. First, they provide new insights into mechanisms whereby enhancers that are targeted by C/EBPB and C/EBPD remain transcriptionally silent during early adipogenesis. Second, they demonstrate that SETD5, a putative histone methyltransferase whose function is poorly understood, plays a structural role in formation of a complex with NCoR-HDAC3 repressor complex that keeps enhancers hypoacetylated and in a "primed" state by restricting recruitment of histone acetyltransferases (HATs) to the enhancers of *Cebpa* and *Pparg* genes. Third, they demonstrate the role of CDC20-APC/C protein degradation axis in regulation of enhancer accessibility.

The study is well-designed, carefully executed and described in sufficient detail.

Comments:

1. The manuscript, while well written, has numerous typographical errors and needs to be carefully proofed.
2. The annotation of the *Pparg* gene locus on Figs. 4C, 5B and 6B showing genome browser snapshots is confusing. Since there is only one *Pparg* gene in mouse, some explanation for there being both a *Pparg1* and *Pparg2* at the bottom of these images would be helpful. Interestingly, the +60 kb enhancer appears to be in a promoter region for the short isoform. Is this region enriched for H4K4me3 marks? If yes, it could be an alternate promoter and not an enhancer.

Reviewer #2 (Remarks to the Author):

In this manuscript, Matsumura et al identifies SETD5 as an important negative regulator of 3T3-L1 adipogenesis. The authors show that SETD5 controls the transition of enhancers from a primed to a fully active state, which is required for transcriptional activation of *Pparg* and *Cebpa* that encode master regulators of late adipocyte differentiation. The authors show that SETD5 is dynamically regulated during early adipogenesis by proteasomal degradation controlled by CDC20 and the APC/C complex. SETD5 is at least partly recruited to primed enhancers by C/EBPb and C/EBPd. While SETD5 does not appear to regulate recruitment of co-repressors to these primed enhancers, it is critical for recruitment of CBP and subsequent full enhancer activation.

This manuscript reports a potentially interesting mechanism for how adipogenic enhancers transition from a primed to an active state. The functional data implicating SETD5 in the control of the adipogenic programme is strong. However, I have some concerns about the robustness of some of the presented data (i.e. number of replicates) and a lack of thorough experimental support for the proposed mechanism. I have detailed some of these concerns below.

Major points

1. The proteomics analyses shown in Fig. 1k are potentially really interesting. However, it is unclear how many replicates were made. Given that no p-values are indicated and changes are assessed only by fold changes, it seems likely that only one replicate was prepared. Since there is also no indication of the number of replicates for the ChIP-seq analyses performed in this manuscript, I suspect these were also only performed once, which makes it very questionable to look at differences between conditions. The proteomics analyses should be validated in a second replicate or at the very least, proteins that appear to be different should be validated by alternative methods, e.g. co-IP or proximity ligation assays. All ChIP-seq data should be performed in at least duplicates followed by proper statistical tests (e.g. DESeq2) to identify differentially regulated regions. Furthermore, more thorough description of the proteomics data is needed. For instance, I assume the bait protein (i.e. SETD5) was identified? If so, how robustly, i.e. how many peptides? Also, how many proteins are identified in total?
2. In Fig. 2A, it appears that there is a striking negative correlation between the H3K27ac levels in the empty control cells vs the SETD5 overexpressing cells at 48 hours. Is this really the case and what would this mean for the function of SETD5? It would help to show the actual H3K27ac levels using more standard heatmaps as the ones used in Fig. 2d.
3. The authors state that CDC20 induces degradation of SETD5 through APC/C, but there is limited evidence of a direct connection between CDC20 and SETD5. CDC20 seems to be required for degradation of SETD5 (Suppl. Fig. 3h), but it is not clear if this effect is through a direct interaction between these proteins, which would be expected given that CDC20 is involved in substrate recognition for the APC/C complex. The proteomics analyses in Fig. 1k do not appear to show that these proteins associate. The authors should investigate whether CDC20 and SETD5 associate, which to this reviewer is a pre-requisite for the proposed mechanism.
4. Fig. 4b shows that ectopically expressed SETD5 binds to H3K27ac-activated enhancers. It would be interesting to relate this binding of SETD5 to the functional importance of SETD5 illustrated in Fig. 2a, i.e. is MDI-induced H3K27ac at these SETD5-bound enhancers also repressed by SETD5 overexpression? This would demonstrate a direct effect of SETD5 on H3K27ac at bound primed enhancers.
5. While it is clear that SETD5 plays an important functional role in repressing adipogenesis, the mechanism underlying this effect is less clear. The authors conclude that proteasomal degradation of SETD5 leads to loss of this protein from primed enhancers resulting in increased HAT recruitment and enhancer activation. However, this does not appear to be fully supported by the provided data and should be explored further. In the SETD5-V5 cells, where SETD5-V5 is completely lost at the tested primed sites at 24h (Fig. 4d), CBP binding is not increased at this timepoint (Fig. 6c). Thus, although SETD5 does appear to restrict CBP binding to chromatin at these primed sites (Fig. 6c), it does not seem to be directly linked to the binding of SETD5 to chromatin indicating an indirect effect (could CBP protein levels be affected? Effect on other histone marks?). Similarly, why does depletion of SETD5 affect binding of CBP to primed enhancers at 48 hours, when there is no (or very little) SETD5 protein and no SETD5 binding to chromatin at this timepoint anyway? This again points to an indirect effect, which likely involves a mechanism operating within the first 24h of adipogenesis, when SETD5 is actually recruited to chromatin at these primed enhancers. Given that this is a key component of the proposed mechanism of how SETD5 controls transition from primed to active enhancers directly on chromatin, the authors should explore this proposed mechanism of SETD5 in more detail.
6. Is the central domain of SETD5 required for adipogenesis also required for recruitment of SETD5 to primed enhancers? The authors have already performed ChIP-qPCR for the WT SETD5-V5 protein and showed that recruitment of SETD5 is at least partly dependent on C/EBP β and C/EBP δ , and it would be interesting to include the mutant in these analyses to determine if this domain is required for

chromatin binding. If it is required for chromatin binding, this would be consistent with the loss of association between SETD5 and the co-repressors shown in the proteomics analyses in Fig. 1k that are recruited to these primed enhancers.

Minor points

1. The poised chromatin state mentioned on p. 4 has mostly been linked to ESCs, which is nicely illustrated in the Fig. 1A, but it might be worth mentioning in the text as well, and also include a short statement saying they are also marked by H3K4me1.
2. Are the changes in mRNA abundance determined by microarray analyses significant, e.g. are the differences shown in Fig. 1b and 1f significant?
3. On p. 8, the authors state that Hi-C has been used to identify functional enhancers of Cebpb. However, promoter capture Hi-C used in the referenced paper does not test functionality, but merely identifies looping between genomic regions. The authors should change their statement to reflect this.
4. Related to point 3 above, was looping between distal enhancers and the promoter of Cebpd also identified in the referenced paper to support that the regions mentioned in the text actually control Cebpd expression? If this is the case, this looping between enhancers and promoters for both Cebpd and Cebpb could be indicated in the Suppl. Fig. 1b to emphasise this point.
5. The last sentence on p. 13 sounds like MDI suppresses ORO, and this reviewer had to read it a few times before it became clear this was not the intended statement of the authors. Maybe consider rephrasing.
6. At the bottom of p. 26 and other places in the discussion, the authors state that SETD5 restricts accessibility of HATs to chromatin. This is too strong in my opinion, since the authors have only investigated one HAT, i.e. CBP.

Reviewer #3 (Remarks to the Author):

In this manuscript, Matsumura et al. tried to address how hypoacetylated "primed" enhancers can transit to hyperacetylated-active state during adipogenesis using mouse 3T3-L1 cell line. The authors show that SETD5 protein levels were transiently increased and rapidly degraded prior to histone acetylation on Cebpa and Pparg genes during adipogenesis. They confirmed that SETD5 forms a complex with NCoR-HDAC3 to prevent enhancer activation. The authors performed SETD5 ChIP-seq during adipogenesis and show that SETD5 localizes on enhancers of Cebpa and Pparg genes before enhancer activation during early adipogenesis. The authors claim that SETD5 delays the induction of adipogenic transcription factors C/EBP α and PPAR γ during adipogenesis by preventing histone acetylation on enhancers. This study makes an interesting connection between the induction of adipogenic TFs and the degradation of SETD5 during adipogenesis in 3T3-L1 cells. However, further functional studies in mice are needed to validate the role of SETD5 and associated NCoR-HDAC3 complex in vivo and to provide novel insights.

Major comments:

1. The authors used mouse 3T3-L1 cells to study the function of the SETD5-containing NCoR-HDAC3 complex during adipogenesis in vitro. Numerous artifacts have been reported in the literature using 3T3-L1 adipogenesis in culture. The authors should use an animal model to validate the role of SETD5 in adipose development in vivo.

2. Figure 1e shows that PPAR γ ligand troglitazone (Tro) treatment fails to rescue adipogenesis in 3T3-L1 cells expressing ectopic SETD5. Since SETD5 overexpression prevents the induction of endogenous Ppar γ expression, the authors need to examine whether ectopic PPAR γ can rescue the adipogenesis defect in SETD5 overexpressing cells.

3. Please use SETD5 antibody to compare the ectopic SETD5 expression levels with those of endogenous SETD5.

4. In Figure 2a, since H3K27ac is highly correlated with gene expression, it is difficult to rule out the possibility that the decreased H3K27ac in SETD5 overexpression cells is due to the secondary effects from the defect of adipogenesis. The authors should confirm the direct effect of SETD5 on adipogenic transcription factor-activated enhancers using C/EBP α or PPAR γ overexpressing 3T3L1 preadipocytes without inducing differentiation.

5. In Figure 3b, please show the expression levels of other subunits in the same complex, at least NCoR2. The authors need to check whether the expression patterns of other subunits are correlated with SETD5 expression during adipogenesis.

Minor comments:

1. Setd5 mRNA levels are decreased upon MDI treatment during early adipogenesis as shown in Figure 1b, but the SETD5 protein levels are increased during the transition phase as shown in Figure 3a. The authors need to explain this inconsistent pattern.

2. In Supplementary Figure 2b, the tables and the genome snapshot need to be aligned.

Responses to reviewer's comments

Reviewer #1: *During cellular differentiation, enhancers are modified by the addition/subtraction of histone marks that determine whether they are poised, primed, or active. A very well studied cell differentiation program is the conversion of preadipocytes into adipocytes. In these studies, Matsumura et al. find that the complex formed between SET domain-containing 5 (SETD5) and the NCoR-HDAC3 co-repressor prevents histone acetylation of enhancers for two master adipogenic regulatory genes Cebpa and Pparg early during adipogenesis. Loss of SETD5 from the complex caused enhancer hyperacetylation, and SETD5 protein levels were transiently increased and rapidly degraded prior to enhancer activation. Induction of the CDC20 co-activator of ubiquitin ligase leads to APC/C mediated degradation of SETD5 during the differentiation transition, which may provide a molecular switch that facilitates adipogenesis.*

These studies are of interest for at least two reasons. First, they provide new insights into mechanisms whereby enhancers that are targeted by C/EBPB and C/EBPD remain transcriptionally silent during early adipogenesis. Second, they demonstrate that SETD5, a putative histone methyltransferase whose function is poorly understood, plays a structural role in formation of a complex with NCoR-HDAC3 repressor complex that keeps enhancers hypoacetylated and in a "primed" state by restricting recruitment of histone acetyltransferases (HATs) to the enhancers of Cebpa and Pparg genes. Third, they demonstrate the role of CDC20-APC/C protein degradation axis in regulation of enhancer accessibility.

Comments:

Comment 1) *The manuscript, while well written, has numerous typographical errors and needs to be carefully proofed.*

Reply: We apologize that the original manuscript had numerous typographical errors. We have carefully proofed the manuscript, and the typographical errors have been corrected.

Comment 2: *The annotation of the Pparg gene locus on Figs. 4C, 5B and 6B showing genome browser snapshots is confusing. Since there is only one Pparg gene in mouse, some explanation for there being both a Pparg1 and Pparg2 at the bottom of these images would be helpful. Interestingly, the +60 kb enhancer appears to be in a promoter region for the short isoform (**Major comment 2-1**). Is this region enriched for H4K4me3 marks? If yes, it could be an alternate promoter and not an enhancer (**Major comment 2-2**).*

Reply to comment 2-1: We appreciate the valuable comment made by the reviewer. As pointed out, mouse has one *Pparg* gene that is transcribed to *Pparg1* and *Pparg2* mRNAs due to alternative transcription start sites (TSSs). *Pparg* +60 kb is a promoter region of *Pparg2* mRNA while this also acts as an enhancer (described in reply to comment 2-2). We have added explanations for this in the revised manuscript and indicated two TSSs in revised Figures 2e, 4c, 5b, 6b, and Supplementary Figures 1b, 1c, 2e.

Revised Fig. 2e Genome browser representation for H3K27ac, C/EBP β , and C/EBP δ on adipogenic genes. 3T3-L1 preadipocytes transduced with empty virus or SETD5 were subjected to ChIP-seq for H3K27ac at the indicated day of differentiation. *Pparg* gene has two transcription start sites (TSS). Tracks 1, 2, and 7 are the same data as Supplementary Fig. 1c. ChIP-seq for C/EBP β and C/EBP δ was from the previously published paper ¹⁹.

Reply to comment 2-2: H3K4me3 mark is not enriched at *Pparg* +60 kb before differentiation induction and becomes enriched at 48 hrs of differentiation (Supplementary Fig. 1d) when *Pparg2* mRNA is started to be transcribed. These results indicate that *Pparg* +60 kb region acts as an enhancer of *Pparg1* until 48 hrs of differentiation and becomes promoter of *Pparg2* after 48 hrs. We have added the following sentences in the revised manuscript. Based recent data base Enhancer Atlas 2.0 (Gao T. and Qian J., Nuc Acid Res, 2020) showed that *Pparg* +60 kb and downstream region is annotated as enhancers in 3T3-L1 cells (Fig. 1 only for reviewers).

“*Pparg* gene uses an alternative transcription start site (+60 kb) to produce *Pparg2* mRNA. H3K4me3 mark is not enriched at *Pparg* +60 kb before differentiation but it becomes enriched by 48 hrs of differentiation (Supplementary Fig. 1d) which is coincident with the start of *Pparg2* transcription. These results indicate that *Pparg* +60 kb region acts as an enhancer of *Pparg1* until 48 hrs of differentiation and then promotes expression of *Pparg2* after 48 hrs.”
(Page 8 lines 1-2 from the bottom to Page 9 lines 1-5)

Fig. 1 only for reviewers

Comparison of H3K27 acetylated regions on *Pparg* gene (upper, from Fig. 2e) and annotated distal enhancers in Enhancer Atlas 2 (lower, <http://enhanceratlas.org/index.php>).

Supplementary Fig. 1d ChIP-qPCR analysis of H3K4me3 on *Pparg* gene during adipogenesis. 3T3-L1 preadipocytes were subjected to ChIP-qPCR analysis using anti-H3K4me3 antibody.

Error bars represent mean \pm SD of three technical replicates.

Reviewer #2: *In this manuscript, Matsumura et al identifies SETD5 as an important negative regulator of 3T3-L1 adipogenesis. The authors show that SETD5 controls the transition of enhancers from a primed to a fully active state, which is required for transcriptional activation of Pparg and Cebpa that encode master regulators of late adipocyte differentiation. The authors show that SETD5 is dynamically regulated during early adipogenesis by proteasomal degradation controlled by CDC20 and the APC/C complex. SETD5 is at least partly recruited to primed enhancers by C/EBP β and C/EBP δ . While SETD5 does not appear to regulate recruitment of co-repressors to these primed enhancers, it is critical for recruitment of CBP and subsequent full enhancer activation.*

This manuscript reports a potentially interesting mechanism for how adipogenic enhancers transition from a primed to an active state. The functional data implicating SETD5 in the control of the adipogenic programme is strong. However, I have some concerns about the robustness of some of the presented data (i.e. number of replicates) and a lack of thorough experimental support for the proposed mechanism. I have detailed some of these concerns below.

Major points:

Comment 1) *The proteomics analyses shown in Fig. 1k are potentially really interesting. However, it is unclear how many replicates were made. Given that no p-values are indicated and changes are assessed only by fold changes, it seems likely that only one replicate was prepared. Since there is also no indication of the number of replicates for the ChIP-seq analyses performed in this manuscript, I suspect these were also only performed once, which makes it very questionable to look at differences between conditions. The proteomics analyses should be validated in a second replicate or at the very least, proteins that appear to be different should be validated by alternative methods, e.g. co-IP or proximity ligation assays (**Major comment 1-1**). All ChIP-seq data should be performed in at least duplicates followed by proper statistical tests (e.g. DESeq2) to identify differentially regulated regions (**Major comment 1-2**).*

*Furthermore, more thorough description of the proteomics data is needed. For instance, I assume the bait protein (i.e. SETD5) was identified? If so, how robustly, i.e. how many peptides? Also, how many proteins are identified in total? (**Major comment 1-3**)*

Reply to comment 1-1:

We appreciate a number of valuable comments made by the reviewer. As pointed out, the data shown in the original manuscript are from the only one best data set from multiple proteomics analyses. To validate the SETD5 interacting proteins, we have performed co-immunoprecipitation analyses of V5-tagged SETD5 followed by immunoblot analyses. This analysis validated the interaction of SETD5 with both NCoR2 and HDAC3 (Supplementary Fig. 1m), ANAPC2 and

ANPAC11 of APC/C subunits (Supplementary Fig. 3f in the revised manuscript).

Supplementary Fig. 1m Co-immunoprecipitation of SETD5 with NCoR2 and HDAC3. Nuclear fraction (Nuc) or whole-cell lysate (WCL) from 3T3-L1 preadipocytes transduced with SETD5-V5 was subjected to immunoprecipitation using anti-V5 antibody followed by immunoblot (IB) analysis using anti-NCoR2, anti-HDAC3, or anti-V5 antibody.

Supplementary Fig. 3f Co-immunoprecipitation of SETD5 with ANAPC2 and ANAPC11. Whole-cell lysate (WCL) from 3T3-L1 preadipocytes transduced with SETD5-V5 at the indicated time of differentiation was subjected to immunoprecipitation using anti-V5 antibody followed by immunoblot (IB) analysis using anti-ANAPC2, anti-ANAPC11, or anti-V5 antibody.

Reply to comment 1-2: As pointed out, we performed another biological replicate of H3K27ac and SETD5-V5 ChIP-seq. Principal component (PC) analyses and scatter plots show the reproducibility of each ChIP-seq in two biological replicates (Supplementary Fig. 2a, 2b, 4f, 4g). Statistical analysis of H3K27ac ChIP-seq using DeSeq2 identified H3K27 acetylated regions that were differentially regulated between empty and SETD5 vector transduced cells at 48 h of differentiation (Supplementary Fig. 2c). The data are consistent with the one in our original manuscript.

Supplementary Fig. 2a, b Principal component (PC) analysis on H3K27ac ChIP-seq peaks in 3T3-L1 preadipocytes transduced with control empty virus and SETD5-V5 (a). Scatter plots showing the reproducibility of H3K27ac ChIP-seq enrichment at individual peaks between two biological replicates (b).

Supplementary Fig. 4f, 4g Principal component (PC) analysis on SETD5-V5 ChIP-seq peaks in 3T3-L1 preadipocytes transduced with SETD5-V5 at the indicated time after MDI induction (f). Scatter plots showing the reproducibility of SETD5-V5 ChIP-seq enrichment at individual peaks between two biological replicates (g).

Supplementary Fig. 2c Statistical analysis of differentially regulated H3K27 acetylated regions using DeSeq2. H3K27 acetylation at 48 hrs was repressed in 3,319 regions (less than 0.5-fold) and induced in 437 regions (more than 2-fold) in SETD5 transduced cells compared to empty virus transduced cells.

Reply to comment 1-3: According to the comment, we added the information of the number of bait protein peptides and identified proteins that interacted with SETD5 protein in Supplementary Table 1.

Accession Number	Molecular Weight	Total Spectrum Count		
		WT	Δ 437-918	Δ SET
SETD5_MOUSE	157 kDa	132	99	134
NCOR2_MOUSE	270 kDa	31	0	45
NCOR1_MOUSE	271 kDa	30	0	31
TBL1X_MOUSE	57 kDa	16	0	18
TBL1R_MOUSE	56 kDa	10	0	12
SKI_MOUSE	80 kDa	5	0	3
HDAC3_MOUSE	48 kDa	5	0	3
ASUN_MOUSE	83 kDa	5	0	4
PLAK_MOUSE	82 kDa	5	0	10
RCC2_MOUSE	56 kDa	7	1	4

PLRG1_MOUSE	57 kDa	8	3	0
PRP4B_MOUSE	117 kDa	5	2	0
THOC4_MOUSE	27 kDa	5	4	0
TRA2B_MOUSE	34 kDa	5	10	0
CDC73_MOUSE	61 kDa	18	16	14
CTR9_MOUSE	133 kDa	29	30	28
PAF1_MOUSE	61 kDa	15	18	13
LEO1_MOUSE	76 kDa	9	14	7
APC1_MOUSE	216 kDa	6	15	8
ANC2_MOUSE	95 kDa	4	7	6

A part of Supplementary Table 1

In addition, the following sentences were added in the text,

“This analysis showed that 509-512 proteins were identified as interacting proteins of WT, and the mutant forms of SETD5 (Supplementary Table 1).” (Page 13, lines 4-6)

“Co-immunoprecipitation analyses confirmed SETD5 interaction with NCoR2 and HDAC3 (Supplementary Fig. 1m).” (Page 13, lines 14-16)

Comment 2) *In Fig. 2A, it appears that there is a striking negative correlation between the H3K27ac levels in the empty control cells vs the SETD5 overexpressing cells at 48 hours. Is this really the case and what would this mean for the function of SETD5? It would help to show the actual H3K27ac levels using more standard heatmaps as the ones used in Fig. 2d.*

Reply to comment 2: Thank you for the valuable comment, yes, this is really the case and this means that SETD5 represses primed enhancer activations. SETD5 inhibits CBP recruitment, which is pivotal for H3K27 acetylation. According to the valuable comment, we have modified Fig. 2a to more standard heatmaps in the revised manuscript. The model presented here is that SETD5 inhibits enhancer activation from primed state via the complex formation with NCoR and HDAC3 during adipogenesis.

Revised Fig. 2a Heatmap representation of H3K27 acetylation during 3T3-L1 adipogenesis. Heatmap shows ChIP-seq data for H3K27ac at a 20-kb region centered on H3K27ac enriched region in 3T3-L1 preadipocytes transduced with empty virus or SETD5 at the indicated time after MDI induction.

Comment 3) *The authors state that CDC20 induces degradation of SETD5 through APC/C, but there is limited evidence of a direct connection between CDC20 and SETD5. CDC20 seems to be required for degradation of SETD5 (Suppl. Fig. 3h), but it is not clear if this effect is through a direct interaction between these proteins, which would be expected given that CDC20 is involved in substrate recognition for the APC/C complex. The proteomics analyses in Fig. 1k do not appear to show that these proteins associate. The authors should investigate whether CDC20 and SETD5 associate, which to this reviewer is a pre-requisite for the proposed mechanism.*

Reply to comment 3: As suggested, we performed co-immunoprecipitation analysis after 21 hr of differentiation induced by MDI cocktail. Proteasome inhibitor MG132 was added at 9 hr of differentiation to inhibit SETD5 degradation. This experiment demonstrated SETD5 co-immunoprecipitated native CDC20 (Supplementary Fig. 3g), which is not contradictory our model that SETD5 is a substrate for CDC20. As commented, in the proteomics analysis in Fig. 1j, CDC20 protein was not co-immunoprecipitated. This is because this proteomics was performed under pre-differentiation condition (i.e. 0 hr) where CDC20 was not expressed yet (Fig. 3b). We thank the reviewer for critical comment.

g
Supplementary Fig. 3g Co-immunoprecipitation of SETD5 with CDC20. 3T3-L1 preadipocytes transduced with SETD5-V5 were induced with MDI and then treated with MG-132 and whole-cell lysate was subjected to immunoprecipitation using anti-V5 antibody followed by immunoblot (IB) analysis using anti-CDC20 or anti-V5 antibody.

Comment 4) *Fig. 4b shows that ectopically expressed SETD5 binds to H3K27ac-activated enhancers. It would be interesting to relate this binding of SETD5 to the functional importance of SETD5 illustrated in Fig.2a, i.e. is MDI-induced H3K27ac at these SETD5-bound enhancers also repressed by SETD5 overexpression? This would demonstrate a direct effect of SETD5 on H3K27ac at bound primed enhancers.*

Reply to comment 4:

We deeply thank the reviewer for this brilliant suggestion. We compared MDI-induced H3K27 acetylated sites with SETD5 binding sites. Of 3,354 “SETD5-repressed H3K27ac sites” (i.e. the sites that H3K27 acetylation was induced by MDI in empty cells, but was not induced in SETD5 transduced cells at 48 hrs differentiation), which includes 2,958 sites in enhancers and 396 sites in promoters (see Fig. 2a), 1,325 sites (40%) were bound by SETD5 at 6 hr of differentiation (Fig. 4b, right pie chart) indicating that SETD5 directly represses H3K27 acetylation of these chromatin. By contrast, 2,029 “SETD5-repressed H3K27 acetylated sites” were not bound by SETD5, suggesting that SETD5 indirectly represses H3K27 acetylation.

Revised Fig. 4b Venn diagram (left) and heatmap (middle) representations of genomic regions occupied by SETD5-V5, H3K4me1, and H3K27ac. Heatmap analysis showing ChIP-seq data for SETD5-V5, H3K4me1, and H3K27ac at a 20-kb region centered on SETD5-V5 binding regions. Pie chart (right) shows SETD5-V5 binding on H3K27 acetylated regions repressed by SETD5.

Comment 5) While it is clear that SETD5 plays an important functional role in repressing adipogenesis, the mechanism underlying this effect is less clear. The authors conclude that proteasomal degradation of SETD5 leads to loss of this protein from primed enhancers resulting in increased HAT recruitment and enhancer activation. However, this does not appear to be fully supported by the provided data and should be explored further. In the SETD5-V5 cells, where SETD5-V5 is completely lost at the tested primed sites at 24h (Fig. 4d), CBP binding is not increased at this timepoint (Fig. 6c). Thus, although SETD5 does appear to restrict CBP binding to chromatin at these primed sites (Fig. 6c), it does not seem to be directly linked to the binding of SETD5 to chromatin indicating an indirect effect (could CBP protein levels be affected? Effect on other histone marks?) (**Major comment 5-1**). Similarly, why does depletion of SETD5 affect binding of CBP to primed enhancers at 48 hours, when there is no (or very little) SETD5 protein and no SETD5 binding to chromatin at this timepoint anyway? (**Major comment 5-2**) This again points to an indirect effect, which likely involves a mechanism operating within the first 24h of adipogenesis, when SETD5 is actually recruited to chromatin at these primed enhancers. Given that this is a key component of the proposed mechanism of how SETD5 controls transition from primed to active enhancers directly on chromatin, the authors should explore this proposed mechanism of SETD5 in more detail.

Reply to Comment 5

Reply to comment 5-1: As pointed out by the reviewer, it would be possible that overexpressed SETD5 indirectly inhibited CBP recruitment to primed enhancers. To answer the criticisms that “Given that this is a key component of the proposed mechanism of how SETD5

controls transition from primed to active enhancers directly on chromatin, the authors should explore this proposed mechanism of SETD5 in more detail”, we performed additional experiments using SETD5 deletion mutant. SETD5 Δ 437-918 which lacks the domain required for the inhibition of adipogenesis (Fig. 1j) and this domain (aa 437-918) is required for the association of SETD5 with NCoR and HDAC3 (Fig. 1k). The new experiments showed that (i) V5-tagged Δ 437-918 SETD5 and WT SETD5 were both recruited to *Cebpa* and *Pparg* enhancers and their recruitment was increased at 6 hr of differentiation compared to before differentiation (0 hr) as demonstrated by V5-ChIP (Supplementary Fig. 4i, also related to comment 6). However, (ii) unlike WT SETD5, Δ 437-918 SETD5 mutant could not inhibit the induction of *Cebpa* and *Pparg* mRNA expression and, (iii) concomitant H3K27 acetylation at the chromatin of these genes (Supplementary Fig. 4j and 4k, respectively). (iv) CBP recruitment to the *Cebpa* and *Pparg* enhancers during 6 and 12 hr of differentiation was inhibited in WT-SETD5 cells while they were not inhibited in Δ 437-918 SETD5 cells compared to those in empty control cells as demonstrated by CBP-ChIP (Fig. 6d), indicating that aa 437-918 are required for restricting CBP recruitment while this domain is not required for the association to the enhancers, whose interactions are through binding to C/EBP β . Because this domain is required for the interaction with NCoR and HDAC3, NCoR-HDAC3 recruitment and CBP recruitment to enhancers are mutually exclusive as illustrated in supplementary Fig. 6b.

Regarding the comments “In the SETD5-V5 cells, where SETD5-V5 is completely lost at the tested primed sites at 24h (Fig. 4d), CBP binding is not increased at this timepoint (Fig. 6c)”, we do not have a clear answer but we are thinking some secondary effect of SETD5 expression, for example, some SETD5 interacting protein(s) may stay with primed enhancers and contribute to restricting CBP recruitment.

Supplementary Fig. 4i ChIP-qPCR analysis wild-type and Δ 437-918 SETD5-V5 on *Cebpa* and *Pparg* genes. 3T3-L1 preadipocytes transduced with wild-type and Δ 437-918 SETD5-V5 were subjected to ChIP-qPCR analysis using anti-V5 antibody at the indicated time after differentiation induction. Data are represented as mean \pm SEM of three independent experiments. Statistical test was performed for comparisons of the group (Tukey’s post hoc comparison). **p<0.01.

j
Supplementary Fig. 4j Transcriptional changes of *Cebpa* and *Pparg* in wild-type and $\Delta 437-918$ SETD5-V5 transduced 3T3-L1 preadipocytes during early adipogenesis. mRNA expression was quantified by qPCR. Cyclophilin mRNA was used as the invariant control. The experiments were performed twice, and the representative one is shown. Error bars represent mean \pm SD of three technical replicates. Statistical test was performed for comparisons of the group (Tukey's post hoc comparison). **p < 0.01.

k
Supplementary Fig. 4k ChIP-qPCR analysis of H3K27ac on *Cebpa* and *Pparg* genes during adipogenesis. 3T3-L1 preadipocytes transduced with wild-type and $\Delta 437-918$ SETD5-V5 were subjected to ChIP-qPCR analysis of H3K27ac at the indicated time after MDI induction of differentiation. The experiments were performed twice, and the representative one is shown. Data are represented as mean \pm SD of three technical replicates. Statistical test was performed for comparisons of the group (Tukey's post hoc comparison). **p < 0.01.

Fig. 6d ChIP-qPCR analyses of CBP on *Cebpa* and *Pparg* genes during the early adipogenesis. 3T3-L1 preadipocytes transduced with control empty virus, wild-type SETD5, or $\Delta 437-918$ mutant were subjected to ChIP-qPCR analysis using anti-CBP antibody. The experiments were performed twice, and the representative one is shown. Error bars represent mean \pm SD of three technical replicates. Statistical test was performed for comparisons of the group (Tukey's post hoc comparison). * $p < 0.05$; ** $p < 0.01$.

Supplementary Fig. 6b Schematic model for regulation of CBP recruitment by SETD5-NCOR-HDAC3 complex. Wild-type SETD5 forms complex with NCoR and HDAC3 and is recruited to primed enhancers via C/EBP transcription factors to restrict CBP recruitment (*left*). The $\Delta 437-918$ mutant SETD5 is recruited to primed enhancers but does not form complex with NCoR and HDAC3 and does not restrict CBP recruitment (*right*).

Reply to comment 5-2:

First of all, we apologize that we did not clearly describe the experimental conditions and brought real confusion. For differentiation, we used two different stimuli: one is a strong MDI full cocktail treatment and the other single dexamethasone (DEX) treatment that is a weak adipogenic stimuli, and we did not clearly explain the condition in each experiment. To avoid confusion, we now performed additional experiments in MDI conditions and the results are summarized below.

(I) Under strong MDI adipogenic cocktail induction,

(i) siRNA mediated knockdown of SETD5 resulted in prematurely early elevation of *Cebpa*

and *Pparg* compared to those in control siRNA cells (i.e., at 24 hr) (Revised Fig. 1g).

(ii) SETD5 knockdown lead to higher elevation of CBP recruitment to *Cebpa* and *Pparg* enhancers at 6 hr and 12 hr of differentiation compared to si-Ctr cells (Fig. 6e); during this 6-12 hr of differentiation, SETD5 protein was stabilized in control cells (Fig. 3a).

(iii) At 24 hr of differentiation, the recruitment of CBP did not differ between SETD5 knockdown and control cells (Fig. 6e); at this time point SETD5 protein was hardly detected due to APC/C and CDC20 mediated proteasomal degradation of SETD5 (Fig. 3 b).

(iv) SETD5 knockdown also resulted in the prematurely early elevation of H3K27ac on *Cebpa* and *Pparg* enhancers at 12 hr and 24 hr of differentiation (Supplementary Fig. 6c) which is consistent with their mRNA levels (revised Fig. 1g).

These time courses are illustrated in Fig. 5f.

Revised Fig. 1g siRNA-mediated knockdown of SETD5 in 3T3-L1 preadipocytes. *Cebpa* and *Pparg* mRNA expression was quantified by qPCR. Cyclophilin mRNA was used as the invariant control. Data are represented as mean \pm SEM of three independent experiments. Statistical test was performed for comparisons of the group (Tukey's post hoc comparison). * $p < 0.05$; ** $p < 0.01$.

Fig. 6e ChIP-qPCR analyses of CBP on *Cebpa* and *Pparg* genes during the early adipogenesis. 3T3-L1 preadipocytes transfected with control siRNA or siRNA targeting to *Setd5* were subjected to ChIP-qPCR analyses using anti-CBP antibody. The experiments were performed twice, and the representative one is shown. Error bars represent mean \pm SD of three technical replicates. Statistical test was performed for comparisons of the group (Tukey's post hoc comparison).

*p<0.05; **p<0.01.

Supplementary Fig. 6c ChIP-qPCR analysis of H3K27ac on *Cebpa* and *Pparg* genes in SETD5-knocked down preadipocytes. 3T3-L1 preadipocytes transfected with control siRNA or siRNA targeting to *Setd5* were subjected to ChIP-qPCR analyses using anti-H3K27ac antibody. The experiments were performed twice, and the representative one is shown. Error bars represent mean \pm SD of three technical replicates. Statistical test was performed for comparisons of the group (Tukey's post hoc comparison). **p<0.01.

II) When dexamethasone (Dex) alone was added, knockdown of SETD5 resulted in premature early expression of both *Cebpa* and *Pparg* (Fig. 1h in the original manuscript) compared to Ctr and differentiated into lipid laden adipocytes as opposed to si-Ctr (Fig. 1h in the revised manuscript). Increased CBP recruitment (Fig. 6e in the original manuscript) and H3K27 acetylation (Fig. 2g) at 48 hr of induction were also associated. However, the time course of each molecule is different from each other between DEX and MDI conditions, and thus it is very misleading. To avoid confusion, unless otherwise clearly stated, most of the experiments were performed under MDI and we omitted the potentially confusing experiments shown in Fig. 1h of the original manuscript.

Original Fig. 1h (omitted in the revised manuscript) siRNA-mediated knockdown of SETD5 in 3T3-L1 preadipocytes. *Cebpa* and *Pparg* mRNA expression was quantified by qPCR. Cyclophilin mRNA was used as the invariant control. Data are represented as mean \pm SEM of three independent experiments. Statistical test was performed for comparisons of the group (Tukey's

post hoc comparison). * $p < 0.05$; ** $p < 0.01$.

Fig. 1h ORO staining of SETD5 knocked-down 3T3-L1 preadipocytes. Preadipocytes were induced with Dex, and ORO staining was performed at day 8 of differentiation.

Original Fig. 6e (omitted in the revised manuscript) ChIP-qPCR analysis of CBP on *Cebpa* and *Pparg* enhancers during adipogenesis. 3T3-L1 preadipocytes transfected with siRNA targeted to *Setd5* were subjected to ChIP-qPCR analysis of CBP at the indicated time after Dex induction of differentiation. The experiments were performed twice, and the representative one is shown. Error bars represent mean \pm SD of three technical replicates. Statistical test was performed for comparisons of the group (Tukey's post hoc comparison). ** $p < 0.01$.

Fig. 2g ChIP-qPCR analysis of H3K27ac on *Cebpa* gene during adipogenesis. 3T3-L1 preadipocytes transfected with siRNA targeted to *Setd5* were subjected to ChIP-qPCR analysis of H3K27ac at the indicated time after Dex induction of differentiation. The experiments were

performed twice, and the representative one is shown. Data are represented as mean \pm SD of three technical replicates. Statistical test was performed for comparisons of the group (Tukey's post hoc comparison). ** $p < 0.01$.

Regarding the criticism “Similarly, why does depletion of SETD5 affect binding of CBP to primed enhancers at 48 hours, when there is no (or very little) SETD5 protein and no SETD5 binding to chromatin at this timepoint anyway?”, this experiment was done under Dex only induction condition and SETD5 remained recruited to primed enhancers after 48 hrs of (Fig. 2a only for reviewers). Under this condition, *Cdc20* mRNA was not induced and SETD5 protein levels did not change (Fig. 2b, c only for reviewers). Showing two different conditions are confusing and misleading, therefore, as described above, we present the CBP ChIP data only from the regular MDI full cocktail condition and thus we deleted the data from under DEX only induction conditions in the revised manuscript.

Fig. 2 only for reviewers

a ChIP-qPCR analysis of SETD5-V5 on *Cebpa* and *Pparg* genes during the early adipogenesis. 3T3-L1 preadipocytes transduced with SETD5-V5 were induced with MDI or Dex and subjected to ChIP-qPCR analysis using anti-V5 antibody. The experiments were performed twice, and the representative one is shown. Data are represented as mean \pm SD of three technical replicates. **b** Expression of mRNAs of *Cebpa*, *Pparg*, and *Cdc20* during the early adipogenesis. 3T3-L1 preadipocytes were induced with MDI or Dex and mRNA expression was quantified by qPCR.

Cyclophilin mRNA was used as the invariant control. The experiments were performed twice, and the representative one is shown. Data are represented as \pm SD of three technical replicates. **c** Immunoblot showing expression of SETD5-V5 during early adipogenesis. 3T3-L1 preadipocytes transduced with SETD5-V5 were induced with MDI or Dex and subjected to immunoblot using anti-V5 antibody. Equal loading of the proteins was confirmed by blotting with anti-actin β antibody.

Comment 6) *Is the central domain of SETD5 required for adipogenesis also required for recruitment of SETD5 to primed enhancers? The authors have already performed ChIP-qPCR for the WT SETD5-V5 protein and showed that recruitment of SETD5 is at least partly dependent on C/EBP β and C/EBP δ , and it would be interesting to include the mutant in these analyses to determine if this domain is required for chromatin binding. If it is required for chromatin binding, this would be consistent with the loss of association between SETD5 and the co-repressors shown in the proteomics analyses in Fig. 1k that are recruited to these primed enhancers.*

Reply: Thank you for the excellent comment. According to the suggestion, we have performed additional experiments. As described in reply to comment 5-1, the Δ 437-918 mutant SETD5 protein was recruited to primed enhancers after differentiation induction similar to wild-type SETD5 protein (Supplemental Fig. 4i).

Minor points:

Comment 1) *The poised chromatin state mentioned on p. 4 has mostly been linked to ESCs, which is nicely illustrated in the Fig. 1A, but it might be worth mentioning in the text as well, and also include a short statement saying they are also marked by H3K4me1.*

Reply: We have added the following sentence in the revised manuscript. We appreciate the reviewer's suggestion.

“Poised enhancers in embryonic stem cells (ESCs) are marked by H3K4me1 and H3K27me3.” (Page 9, lines 10-11)

Comment 2) *Are the changes in mRNA abundance determined by microarray analyses significant, e.g. are the differences shown in Fig. 1b and 1f significant?*

Reply: As for Fig. 1f, we have performed multiple qPCR analyses from biological replicates and obtained similar data; induction of adipogenic *Cebpa*, *Pparg*, and *Cd36* mRNA are inhibited in SETD5 overexpressing 3T3-L1 preadipocytes (Fig. 3 only for reviewers). The representative

RNA was subjected to the microarray analyses shown in Fig. 1e. As for Fig. 1b, the same batch of Wnt3a used in the original Fig. 1b is not available, and we could not perform statistical analysis in multiple experiments. Because SETD5 regulation at the protein level is more important than at the transcription level in this manuscript, we would like to omit original Fig. 1b (related to minor comment 1 from the reviewer 3).

Fig. 3 only for reviewers

Expression of mRNAs of *Cebpa*, *Pparg*, and *Cd36* during the early adipogenesis. 3T3-L1 preadipocytes transduced with empty virus or SETD5 were induced with MDI and mRNA expression was quantified by qPCR. Cyclophilin mRNA was used as the invariant control. The experiments were performed three times, and the representative one is shown. Data are represented as \pm SD of three technical replicates. Statistical test was performed for comparisons of the group (Tukey's post hoc comparison). ** $p < 0.01$.

Comment 3) On p. 8, the authors state that Hi-C has been used to identify functional enhancers of *Cebpb*. However, promoter capture Hi-C used in the referenced paper does not test functionality, but merely identifies looping between genomic regions. The authors should change their statement to reflect this.

Reply: As pointed out by the reviewer, the Hi-C represents looping between genomic regions but does not identify functionality of enhancers. Accordingly, we have changed our statement in the revised manuscript as follows.

“Mandrup and her colleagues, using the Hi-C technique, previously identified looping between *Cebpb* promoter and regions located at +77 kb and +88 kb³². As shown in Supplementary Fig. 1b, we observed that these *Cebpb* distal regions (e.g. +77 kb and +88 kb) and also *Cebpd* distal regions (e.g. +77 kb and +88 kb) displayed active enhancer signatures marked by the presence of both H3K4me1 (track 3) and H3K27ac (track 4) which was coupled with a paucity of H3K27me3 (track 2) in preadipocytes before differentiation (i.e. 0 h) indicating that these enhancers are already active before differentiation.” (Page 8, lines 4-11)

Comment 4) Related to point 3 above, was looping between distal enhancers and the promoter of *Cebpd* also identified in the referenced paper to support that the regions mentioned in the text actually control *Cebpd* expression? If this is the case, this looping between enhancers and promoters for both *Cebpd* and *Cebpb* could be indicated in the Suppl. Fig. 1b to emphasise this point.

Reply: Hi-C by Mandrup group used HindIII enzyme, a 6 bp cutter, for library preparation and identified looping between enhancers and promoter on *Cebpb* (Fig. 7D in Siersbæk et al., *Mol Cell*, 2017), but looping on *Cebpd* was not shown. Information in the recent data base Enhancer Atlas 2.0 (Gao T. and Qian J., *Nuc Acid Res*, 2020) showed that both *Cebpb* and *Cebpd* distal regions are annotated as enhancers in 3T3-L1 cells (Fig. 4 only for reviewers). Although we have not published it yet, we have performed Hi-C analysis using MboI enzyme, a 4 bp cutter, in 3T3-L1 preadipocytes. Our data show the interaction of TSS region and the putative enhancers regions: “the looping between enhancers and promoters for both *Cebpd* and *Cebpb*” (Fig. 5 only for reviewers).

Fig. 7D in Siersbæk et al., *Mol Cell* (2017)

WashU screenshot of *Cebpb* and a connected enhancer. Binding of corepressors and coactivators to the enhancer as determined by ChIP-seq (bottom right), promoter-enhancer chromatin loop formation as determined by PCHi-C (top), and *Cebpb* expression as determined by mRNA-seq (bottom left) in preadipocytes (i.e., prior to induction of differentiation) and 4 hr after stimulation with the adipogenic hormone cocktail are shown. The height of the loop as well as the thickness of the loop line indicates the strength of the promoter-enhancer interaction (top). Fold changes

for interaction strength, corepressor/coactivator binding, and expression level are indicated in vertical orientation.

Fig. 4 only for reviewers

Distal enhancers *Cebpb* (upper) and *Cebpd* (lower) genes in 3T3-L1 cells annotated by Enhancer Atlas 2 (<http://enhanceratlas.org/index.php>).

Fig. 5 only for reviewers

Hi-C shows interaction of promoter and enhancers on *Cebpb* and *Cebpd* genes in 3T3-L1 cells. Virtua 4C (*upper*) shows interaction of promoter and distal enhancers (one vs all) on *Cebpb* (*left*) and *Cebpd* (*right*) genes. The position of viewpoint is indicated by the anchor signs. Contact map (*middle*) shows all versus all chromosomal contacts at 2-kb resolution. Darker color indicates stronger contacts. Histone modifications on *Cebpb* and *Cebpd* genes are shown for a comparison (*lower*, from Supplementary Fig. 1b).

Comment 5) *The last sentence on p. 13 sounds like MDI suppresses ORO, and this reviewer had to read it a few times before it became clear this was not the intended statement of the authors. Maybe consider rephrasing.*

Reply: We apologize that it was not clearly written. We have corrected the sentence as follows.

“First, we depleted NCoR2 by siRNA in SETD5-transduced preadipocytes and showed that the suppression of lipid accumulation by SETD5 overexpression was reversed by NCoR2 depletion (Fig. 1k, l).” (Page 14, lines 9-11 in the revised manuscript)

Comment 6) *At the bottom of p. 26 and other places in the discussion, the authors state that SETD5 restricts accessibility of HATs to chromatin. This is too strong in my opinion, since the authors have only investigated one HAT, i.e. CBP.*

Reply: As pointed out by the reviewer, we have modified the sentences to tone down our statement as follows.

- p. 28, “These results indicate that SETD5-NCoR-HDAC3 complex restricts accessibility of CBP, one of HATs, to primed enhancers, thereby regulating the time course of enhancer activation. When SETD5 protein diminishes via proteasomal degradation and consequently its recruitment to primed enhancers declines, CBP recruitment to primed enhancers increases to robustly drive master regulator gene expression and the adipogenic differentiation program (Fig. 6f).”
- p. 30, “SETD5 in an NCoR-HDAC3 complex restricts accessibility of CBP to enhancer chromatin but when SETD5 is lost from this complex via proteasomal degradation requiring the E3 ubiquitin ligase APC/C complex, associated or recruited CBP then hyperacetylates enhancers leading to subsequent gene activation.”
- p. 31, “In early differentiation (until 6-12 hrs), both NCoR-HDAC3-SETD5 co-repressor complex and CBP are tethered to primed enhancers where SETD5 restricts accessibility of CBP to enhancer chromatin and keeps histones hypoacetylated and held in a primed state.”

Reviewer #3: *In this manuscript, Matsumura et al. tried to address how hypoacetylated “primed” enhancers can transit to hyperacetylated-active state during adipogenesis using mouse 3T3-L1 cell line. The authors show that SETD5 protein levels were transiently increased and rapidly degraded prior to histone acetylation on Cebpa and Pparγ genes during adipogenesis. They confirmed that SETD5 forms a complex with NCoR-HDAC3 to prevent enhancer activation. The authors performed SETD5 ChIP-seq during adipogenesis and show that SETD5 localizes on enhancers of Cebpa and Pparγ genes before enhancer activation during early adipogenesis. The authors claim that SETD5 delays the induction of adipogenic transcription factors C/EBPα and PPARγ during adipogenesis by preventing histone acetylation on enhancers. This study makes an interesting connection between the induction of adipogenic TFs and the degradation of SETD5 during adipogenesis in 3T3-L1 cells. However, further functional studies in mice are needed to validate the role of SETD5 and associated NCoR-HDAC3 complex in vivo and to provide novel insights.*

Major comments:

Comment 1) *The authors used mouse 3T3-L1 cells to study the function of the SETD5-containing NCoR-HDAC3 complex during adipogenesis in vitro. Numerous artifacts have been reported in the literature using 3T3-L1 adipogenesis in culture. The authors should use an animal model to validate the role of SETD5 in adipose development in vivo.*

Reply: As pointed out by the reviewer, we examined the role of SETD5 in adipose development in a transplant experiment. Transplantation of control 3T3-L1 preadipocytes into nude mice generated perilipin-1 positive adipocytes, while transplantation of 3T3-L1 preadipocytes overexpressing SETD5 did not generate adipocytes (revised Fig. 7a, b, c, d).

Fig. 7 SETD5 inhibits adipogenesis in vivo

a Transplantation of 3T3-L1 preadipocytes into nude mice. 3T3-L1 preadipocytes transduced with control empty virus or SETD5-V5 were cultured for two days with MDI cocktail. Subsequently, harvested cells were implanted with HydroMatrix and injected into the subcutaneous region on the upper or lower back of nude mice. **b, c** Two weeks after transplantation, mice were dissected, and isolated transplanted fats were subjected to haematoxylin and eosin (H&E) staining (**b**) and immunohistochemistry using anti-perilipin-1 antibody (**c**). Representative images of transplants from the lower back are shown. **d** Quantification of perilipin-1 positive adipocytes. Error bars represent mean \pm SE in the section of transplants from empty virus transduced (n=5) or SETD5 transduced (n=3) preadipocytes (**p < 0.01, Student's t-test).

Comment 2) Figure 1e shows that PPAR γ ligand troglitazone (Tro) treatment fails to rescue adipogenesis in 3T3-L1 cells expressing ectopic SETD5. Since SETD5 overexpression prevents the induction of endogenous Ppar γ expression, the authors need to examine whether ectopic PPAR γ can rescue the adipogenesis defect in SETD5 overexpressing cells.

Reply: As suggested by the reviewer, we established 3T3-L1 preadipocytes expressing both FLAG-tagged SETD5 and human PPAR γ 1 or PPAR γ 2 and examine lipid accumulation after differentiation induction. Revised Supplementary Fig. 1i and 1j show that ectopic expression of human PPAR γ 1 or PPAR γ 2 fully restored lipid accumulation in 3T3-L1 cells expressing SETD5.

Supplementary Fig. 1i, j Retroviral expression of FLAG-tagged SETD5 and human PPAR γ 1 or PPAR γ 2. Whole-cell lysate from 3T3-L1 preadipocytes transduced with the indicated retroviruses was subjected to immunoblot (IB) analysis (*upper*). Equal loading of the proteins was confirmed by blotting with anti-actin β antibody. mRNA expression of human Pparg was quantified by qPCR (*lower*). Cyclophilin mRNA was used as the invariant control. Error bars represent mean \pm SD of three technical replicates. j ORO staining of 3T3-L1 preadipocytes transduced with empty virus or FLAG-tagged SETD5 or together with human PPAR γ 1 or PPAR γ 2. Preadipocytes were induced with MDI mixture and stained on day 8 of differentiation.

Comment 3) Please use SETD5 antibody to compare the ectopic SETD5 expression levels with those of endogenous SETD5.

Reply: As suggested by the reviewer, we compared the ectopic SETD5 expression levels with those of endogenous SETD5 by immunoblot analysis. Fig. 3 only for reviewers shows expression of endogenous and ectopic SETD5 in 3T3-L1 preadipocytes transduced with SETD5 or control empty virus.

Fig. 6 for reviewers. Retroviral expression of SET D5 in 3T3-L1 preadipocytes.

Immunoblot analysis showing expression of endogenous and ectopic SETD5 in 3T3-L1 preadipocytes transduced with SETD5 or control empty virus. Equal loading of the protein was confirmed by blotting with anti-actin β antibody.

Comment 4) In Figure 2a, since H3K27ac is highly correlated with gene expression, it is difficult to rule out the possibility that the decreased H3K27ac in SETD5 overexpression cells is due to the secondary effects from the defect of adipogenesis. The authors should confirm the direct effect of SETD5 on adipogenic transcription factor-activated enhancers using C/EBP α or PPAR γ overexpressing 3T3L1 preadipocytes without inducing differentiation.

Reply: As suggested by the reviewer, we retrovirally transduced C/EBP α or empty vector into 3T3-L1 preadipocytes expressing either empty or V5-tagged SETD5 and examined H3K27 acetylation levels on chromatin of *Cebpa* and *Pparg* enhancers (Supplementary Fig. 2f) without using differentiation induction reagent. In empty 3T3-L1 preadipocytes, C/EBP α transduction facilitated H3K27 acetylation on chromatin of these enhancers (Revised Supplementary Fig. 2g). By contrast, in SETD5 transduced preadipocytes, C/EBP α transduction did not induce H3K27 acetylation on these enhancers (Supplementary Fig. 2g in revised version). We appreciate the valuable suggestion made by the reviewer.

Supplementary Fig. 2f Retroviral expression of FLAG-tagged SETD5 and C/EBP α . Whole-cell lysate from 3T3-L1 preadipocytes transduced with the indicated retroviruses was subjected to immunoblot (IB) analysis (*left*). Equal loading of the proteins was confirmed by blotting with anti-actin β antibody. mRNA expression of retroviral *Cebpa* was quantified by qPCR (*right*). Error bars represent mean \pm SD of three technical replicates.

Supplementary Fig. 2g ChIP-qPCR analysis of H3K27ac on *Cebpa* and *Pparg* genes in 3T3-L1 preadipocytes transduced with SETD5 and C/EBP α . 3T3-L1 preadipocytes transduced with the indicated retroviruses were subjected to ChIP-qPCR analysis of H3K27ac before differentiation induction. The experiments were performed twice, and the representative one is shown. Error bars represent mean \pm SD of three technical replicates. Statistical test was performed for comparisons of the group (Tukey's post hoc comparison). ** $p < 0.01$.

Comment 5) In Figure 3b, please show the expression levels of other subunits in the same complex, at least NCoR2. The authors need to check whether the expression patterns of other subunits are correlated with SETD5 expression during adipogenesis.

Reply: As suggested by the reviewer, we performed immunoblot analysis of NCoR2 and HDAC3 during adipogenesis (Figure 3b in revised version). Protein levels of NCoR2 and HDAC3 did not change during early adipogenesis while SETD5 protein was reduced between 15 and 24 hr of

differentiation as CDC20 protein was increased. These data show the SETD5 protein specific regulation in the repressor complex.

Revised Fig. 3b Immunoblot showing expression of SETD5-V5, NCoR2, HDAC3, and CDC20 in SETD5-transduced 3T3-L1 preadipocytes. Equal loading of the proteins was confirmed by blotting with anti-actin β antibody. The graph shows band intensity of SETD5-V5 and CDC20 at the indicated time after MDI induction. The experiments were performed three times, and the representative one is shown.

Minor comments:

Comment 1) Setd5 mRNA levels are decreased upon MDI treatment during early adipogenesis as shown in Figure 1b, but the SETD5 protein levels are increased during the transition phase as shown in Figure 3a. The authors need to explain this inconsistent pattern.

Reply: Related to the criticism made by reviewer 2, we are unable to perform additional experiments to validate mRNA levels because the same batch of Wnt3a reagent is currently not available. This data is from our transcriptome data base previously published (*Mol Cell Biol*, Wakabayashi K et al, 2009; *Proc Natl Acad Sci U S A*, Okamua M et al, 2009). Regarding the question, we do not have clear answers. Since protein levels are more important than mRNA levels in this paper and this data is confusing to readers, we eliminated this figure together with related heat map shown in Supplementary Fig. 1f.

Comment 2) In Supplementary Figure 2b, the tables and the genome snapshot need to be aligned.

Reply: We really appreciate careful check made by the reviewer. We aligned them carefully and the corrected version is now shown as Supplementary Fig. 2e in the revised manuscript. Panel was moved to **e** from **b** due to the addition of new data.

Original Supplementary Fig. 2b

Revised Supplementary Fig. 2e

REVIEWERS' COMMENTS

Reviewer #2 (Remarks to the Author):

I have now carefully gone through the revised manuscript, and the authors have addressed all my concerns related to the originally submitted manuscript. The authors have put in a lot of effort to improve the manuscript, and I think it is suitable for publication in its current form.

I have two small suggestions for the authors to consider.

1) I would suggest including Fig. 3 only for reviewers in the manuscript. This supports the current Fig. 1e, which is only one replicate. If they chose to do so, I would suggest combining all three biological replicates and do the statistics on the combined biological replicates rather than technical replicates.

2) I would suggest including Fig. 4 only for reviewers in the supplemental section of the manuscript. This supports the notion that the highlighted regions near both the CEBPd and CEBPb genes are in fact enhancers involved in regulating these genes. Maybe even consider including a similar figure for the PPARg and CEBPa genes.

Reviewer #3 (Remarks to the Author):

In their revised manuscript, Matsumura et al have included new experimental data to address reviewers' concerns on the first version. The authors revised Figure 3b to show protein levels of other subunits of the NCoR-HDAC3 corepressor complex in the early phase of adipogenesis. They also investigated the role of SETD5 in adipogenesis by transplanting 3T3-L1 cells overexpressing SETD5 into nude mice (see the new Figure 7). Overall, the revised manuscript provides new insights into the role of SETD5 and associated NCoR-HDAC3 corepressor complex in regulating enhancer activation in the early phase of adipogenesis. However, I still have 2 concerns:

1. The transplantation of 3T3-L1 cells overexpressing SETD5 into nude mice is not an in vivo experiment. It does not prove that SETD5 negatively regulates adipogenesis in vivo.

2. The authors need to change "Spatiotemporal dynamics" to "Temporal dynamics" in the manuscript title. There is no evidence to support spatial dynamics of SETD5 using the in vitro cell culture system.

Responses to reviewer's comments

Reviewer #2: I have now carefully gone through the revised manuscript, and the authors have addressed all my concerns related to the originally submitted manuscript. The authors have put in a lot of effort to improve the manuscript, and I think it is suitable for publication in its current form.

I have two small suggestions for the authors to consider.

Comments:

Comment 1: I would suggest including Fig. 3 only for reviewers in the manuscript. This supports the current Fig. 1e, which is only one replicate. If they chose to do so, I would suggest combining all three biological replicates and do the statistics on the combined biological replicates rather than technical replicates.

Reply: We appreciate kind comment and constructive suggestions made by the reviewer #2. According to the comment, we added qPCR data of combined all three biological replicates with the statistics in Supplementary Figure 1m of the revised manuscript.

Supplementary Figure 1m Expression of mRNA levels of *Cebpa* and *Pparg* during the early adipogenesis. 3T3-L1 preadipocytes transduced with empty or SETD5 retrovirus were induced for differentiation with MDI. mRNA levels at the indicated time were quantified by qPCR. Data are mean \pm SEM of three independent experiments. One-way ANOVA with Tukey's multiple comparisons test. ** $p < 0.01$.

Comment 2: I would suggest including Fig. 4 only for reviewers in the supplemental section of the manuscript. This supports the notion that the highlighted regions near both the *CEBPd* and *CEBPb* genes are in fact enhancers involved in regulating these genes. Maybe even consider

including a similar figure for the *PPARG* and *CEBPA* genes.

Reply: This is an excellent suggestion and according to the comment, we added Enhancer Atlas 2.0 data of *Cebpb*, *Cebpd*, *Cebpa*, and *Pparg* genes as in Supplementary Figure 1c and e of the revised manuscript.

Supplementary Figure 1c, e Enhancer ATLAS 2.0 genome browser representation of distal enhancers of *Cebpb* and *Cebpd* genes (c) and *Cebpa* and *Pparg* genes (e) in 3T3-L1 cells. EP300, E1A-associated protein p300; DHS, Dnase I hypersensitive site; TF, transcription factor; POL2, RNA polymerase II.

Reviewer #3: *In their revised manuscript, Matsumura et al have included new experimental data to address reviewers' concerns on the first version. The authors revised Figure 3b to show protein levels of other subunits of the NCoR-HDAC3 corepressor complex in the early phase of adipogenesis. They also investigated the role of SETD5 in adipogenesis by transplanting 3T3-L1 cells overexpressing SETD5 into nude mice (see the new Figure 7). Overall, the revised manuscript provides new insights into the role of SETD5 and associated NCoR-HDAC3 corepressor complex in regulating enhancer activation in the early phase of adipogenesis. However, I still have 2 concerns:*

Comment 1: *The transplantation of 3T3-L1 cells overexpressing SETD5 into nude mice is not an in vivo experiment. It does not prove that SETD5 negatively regulates adipogenesis in vivo.*

Reply: We thank the reviewer#3 for the careful reading and valuable comments. Although this reviewer commented that the 3T3L1 transplantation study is not “in vivo” experiment, we respectfully disagree with this interpretation of the data. While it is true that the transplanted cells come from tissue culture, once they are implanted into the nude mice, they must rely on the host *in vivo* physiological systems to develop real phenotypic traits expected for normal adipose tissue. In this physiologically relevant setting in a live animal, over expression of SETD5 has a clear effect on differentiation of the transplanted cells.

Comment 2: *The authors need to change “Spatiotemporal dynamics” to “Temporal dynamics” in the manuscript title. There is no evidence to support spatial dynamics of SETD5 using the in vitro cell culture system.*

Reply: We demonstrated “spatial dynamics” of SETD5-containing NCoR-HDAC3 complex in the nucleus of 3T3-L1 preadipocytes by investigating the genome-wide SETD5 binding sites by ChIP-seq and the enhancer-promoter proximity of *Pparg* and *Cebpa* master regulator genes as well as *Cebpb* and *Cebpd* by Hi-C (*Mol Cell*, Siersbaek R, et al. 2017 and our data) and the comprehensive Enhancer Atlas 2.0 database (*Nuc Acids Res*, Gao T and Qian J, 2020) in the revised manuscript in response to the comments made by the Reveiwer#2. The new data are now shown in the **Supplementary Figure 1c and e** in the 2nd version of revised manuscript. Therefore, we hope the editor will agree that the presented data in this 2nd revision are sufficient to support the title “Spatiotemporal dynamics of SETD5-containing NCoR-HDAC3 complex determines enhancer activation for adipogenesis.”